# Effect of whole-body vibration combined with exercise therapy on jump-landing stability after ACL reconstruction: A randomized controlled trial

**Roya Khanmohammadi** [1]*, **Zahra Rostampour** [2]

**1** Assistant Professor, PhD, PT, Iranian Center of Excellence in Physiotherapy, Rehabilitation Research Center, Department of Physiotherapy, School of Rehabilitation Sciences, Iran University of Medical Sciences, Tehran, Iran, **2** MSc, PT, Department of Physiotherapy, School of Rehabilitation, Tehran University of Medical Sciences, Tehran, Iran

* dr.rkhanmohammadi96@gmail.com

## Abstract

This study examined whether incorporating whole-body vibration (WBV) into routine exercises enhances dynamic postural stability during jump-landing in athletes following anterior cruciate ligament reconstruction (ACLR), compared to routine exercises alone. Additionally, the study evaluated the effects of WBV on dynamic balance using the Y-Balance Test (YBT) and on physical performance using the Limb Symmetry Index (LSI) derived from the 6-meter timed hop test. In this randomized, single-blind, controlled trial, thirty participants were assigned to either a WBV plus exercise group or an exercise-only group. Assessments were conducted before and after the intervention. Primary outcomes included the Stability Index (SI) and Time to Stabilization (TTS). Secondary outcomes included the YBT and the LSI. In the WBV group, antero-posterior (AP) TTS decreased significantly ($P = 0.001$, $\eta^2 = 0.568$), indicating a large effect, while the control group showed no significant change ($P = 0.138$, $\eta^2 = 0.150$). A similar pattern was found for resultant TTS: the intervention group demonstrated a significant decrease ($P = 0.001$, $\eta^2 = 0.533$), whereas the control group did not ($P = 0.145$, $\eta^2 = 0.146$). These findings indicate that TTS decreased significantly only in the WBV group, both in the AP direction and in the combined (resultant) measure. For the YBT ($P \leq 0.016$, $\eta^2 \leq 0.542$) and LSI ($P < 0.001$, $\eta^2 = 0.610$), both groups demonstrated significant improvements from pre- to post-intervention, with no significant main effects of group or time × group interactions, indicating comparable changes over time. Overall, the study showed that adding WBV to routine exercises produced selective benefits, with additional improvements observed only in TTS during the jump-landing task. In contrast, both groups demonstrated comparable improvements in YBT, and LSI, with no between-group differences. These findings indicate that WBV did not provide added benefit beyond exercise alone for

**Data availability statement:** All relevant data are within the manuscript and its Supporting Information files.

**Funding:** The author(s) received no specific funding for this work.

**Competing interests:** NO authors have competing interests.

these latter outcomes. **Trial registration number**: Iranian Registry of Clinical Trials (IRCT20220226054122N1) Registered on 01/03/2022 (https://irct.behdasht.gov.ir/trial/62172).

---

## 1. Introduction

Tearing the anterior cruciate ligament (ACL) is a common sports injury. Despite ACL reconstruction (ACLR), long-term consequences persist, including high re-injury risk [1], limited return to sport [2], and potential development of knee osteoarthritis [3]. Around two-thirds of athletes do not return to their previous activity level one year after reconstruction [2], and only 65% resume their pre-injury exercise within ~3.5 years [4]. After 7 years, just 36% continue their main sport [5], with up to 29% risk of re-injuring the same or opposite ligament [6].

Several studies have demonstrated that even after ligament reconstruction and rehabilitation, athletes continue to exhibit impaired postural stability [7–9]. Research also shows that postural control remains compromised compared to healthy individuals even two years after surgery [10,11], highlighting the persistence of these deficits over time. Importantly, impaired postural stability is a well-established predictor of future ligament injury [12]. Moreover, evidence indicates that these neuromuscular deficits may become even more pronounced under fatigue, further compromising postural control and increasing the likelihood of re-injury in individuals after ACLR [13]. Persistent fear of movement (kinesiophobia) is another factor that may hinder return-to-sport and limit functional recovery after ACLR. Meta-analyses indicate that psychosocial interventions can reduce kinesiophobia, which remains a major barrier even after surgery [14].

Together, these findings suggest that persistent deficits in postural control and unresolved kinesiophobia are key contributors to limited functional recovery and increased vulnerability to re-injury. This highlights the importance of incorporating targeted interventions into rehabilitation programs to enhance both postural stabilityand patient confidence, thereby reducing the risk of subsequent injury.

Whole-body vibration (WBV) has emerged as a rehabilitation modality with growing application in musculoskeletal and sports medicine. Several investigations have reported that WBV can facilitate neuromuscular adaptations and enhance postural stability in individuals following ACLR [15–18]. Specifically, Moezy et al. and Berschin et al. reported superior improvements compared to conventional exercise [15,16], while Fu et al. and Pistone et al. observed additional benefits when WBV was integrated into standard multi-session rehabilitation programs [17,18]. In contrast, da Costa et al. found no significant effects of an acute, single-session WBV protocol on postural stability [19]. Collectively, these studies show inconsistent results, highlighting that the effectiveness of WBV in ACL rehabilitation remains equivocal and may depend on the intervention dose and duration.

Beyond the heterogeneity of findings, an important methodological limitation in the current literature on post-ACL rehabilitation is the predominant reliance on relatively simple standing tasks, such as maintaining the center of gravity (COG) on a stable

or mildly perturbed surface (e.g., Biodex platform). While these assessments provide useful information on static postural control, they have limited ecological validity and fail to capture the complex demands of dynamic stability required during sport-specific movements. Such balance tasks on low Biodex stability levels may not fully the multi-planar, high-velocity neuromuscular demands of sport and could therefore underestimate residual deficits following ACLR [11,20].

In contrast, dynamic activities such as jump-landing better reflect the functional challenges encountered during athletic participation, particularly given that the majority of ACL injuries occur under such conditions [11,21,22]. Abnormal landing kinematics and kinetics have been shown to predict re-injury risk in rehabilitated athletes [12], and the ability to rapidly regain stability after landing is recognized as a critical determinant for safe return to sport, making it a pivotal consideration in return-to-sport decision-making [21,23]. Addressing these limitations, the present study evaluates the effects of whole-body vibration (WBV) on dynamic postural stability during jump-landing, providing a functionally relevant assessment of its role in post-ACL rehabilitation.

Dynamic postural stability can be quantitatively assessed using the stability index (SI) and time to stabilization (TTS), both derived from jump-landing tasks. TTS has been shown to detect deficits in dynamic stability among individuals with ACL injuries and ankle instability compared to healthy controls [11,24–26]. Notably, prolonged TTS during backward single-leg jump-landing has been associated with a threefold increase in the risk of future ACL rupture [27]. The SI demonstrates higher repeatability (ICC: 0.96) and accuracy (Standard Error of Measure: 0.03) than TTS [20], and thus, employing both metrics provides a comprehensive evaluation of dynamic postural control.

Despite the increasing integration of WBV into sports rehabilitation, particularly for ACLR, its effects on dynamic postural stability during sport-specific tasks such as jump-landing remain largely unexplored. Preliminary evidence in healthy populations suggests that WBV may reduce ground reaction forces (GRFs) [28]and TTS [29] during jump-landing, however, its effects in post-ACL individuals remain unclear. This gap highlights the need to examine WBV as an adjunct to routine rehabilitation for improving dynamic postural stability in this population.

The primary aim of the present study is to determine whether incorporating WBV into routine rehabilitation exercises enhances dynamic stability during jump-landing in athletes following ACLR, compared to exercises alone. Secondary objectives include evaluating its effects on dynamic balance and physical performance using clinical assessments. We hypothesize that while routine exercises will improve SI, TTS, and functional outcomes, the addition of WBV will further amplify these effects, resulting in significantly greater improvements in dynamic stability, balance, and overall performance.

## 2. Method

### 2.1. Study design

This study was a randomized, single-blinded, controlled trial. Participants were assigned to either a group receiving WBV combined with routine exercises or a group performing routine exercises alone. Outcome measures assessed both before and after the treatment. Post-tests were performed within 24–48 hours after completion of treatment. Primary outcome measures included the stability index and the time to stabilizationin the anteroposterior (AP), mediolateral (ML), and vertical (V) directions, as well as the dynamic postural stability index (DPSI) and resultant vector time to stabilization (RVTTS) during diagonal jump landing. Secondary outcomes encompassed the Y-balance test (YBT) in the anterior, posteromedial (PM), and posterolateral (PL) directions, as well as the limb symmetry index (LSI) in the 6-meter timed hop (6MTH) test. Both groups received treatment in a laboratory setting. This study was registered as a clinical trial in the Iranian Registry of Clinical Trials (IRCT20220226054122N1) on 01/03/2022. The subjects were recruited between April 3, 2022, and August 21, 2023.

### 2.2. Participants

30 athletes who had undergone ACLR were included in the study (Table 1). A flow diagram based on the CONSORT statement illustrates the participants' progress from enrollment to analysis (Fig 1). All experimental procedures involving

**Table 1. The demographical and clinical characteristics of subjects at baseline.**

| Characteristics | Intervention (N = 15) | | Control (N = 15) | | P Value |
|---|---|---|---|---|---|
| | Mean | SD | Mean | SD | |
| Age (years) | 31.73 | 4.82 | 32.47 | 5.62 | 0.70 |
| Height (cm) | 176.00 | 9.61 | 180.33 | 4.24 | 0.13 |
| Weight (kg) | 76.13 | 10.84 | 81.67 | 11.31 | 0.18 |
| Time Since Injury (months) | 19.80 | 9.12 | 16.13 | 4.49 | 0.18 |
| Time Since Operation (months) | 13.47 | 7.17 | 10.00 | 2.04 | 0.09 |
| Physical Activity (hours/week) | 10.00 | 3.21 | 10.53 | 4.02 | 0.69 |
| Marx Physical Activity Level (0–16) | 9.47 | 1.36 | 10.53 | 2.83 | 0.20 |
| Gender (F/M) | 4/11 | | 3/12 | | 0.67 |
| Operated Limb (L/R) | 9/6 | | 8/7 | | 0.71 |

F= Female; M= Male; L= Left; R= Right.

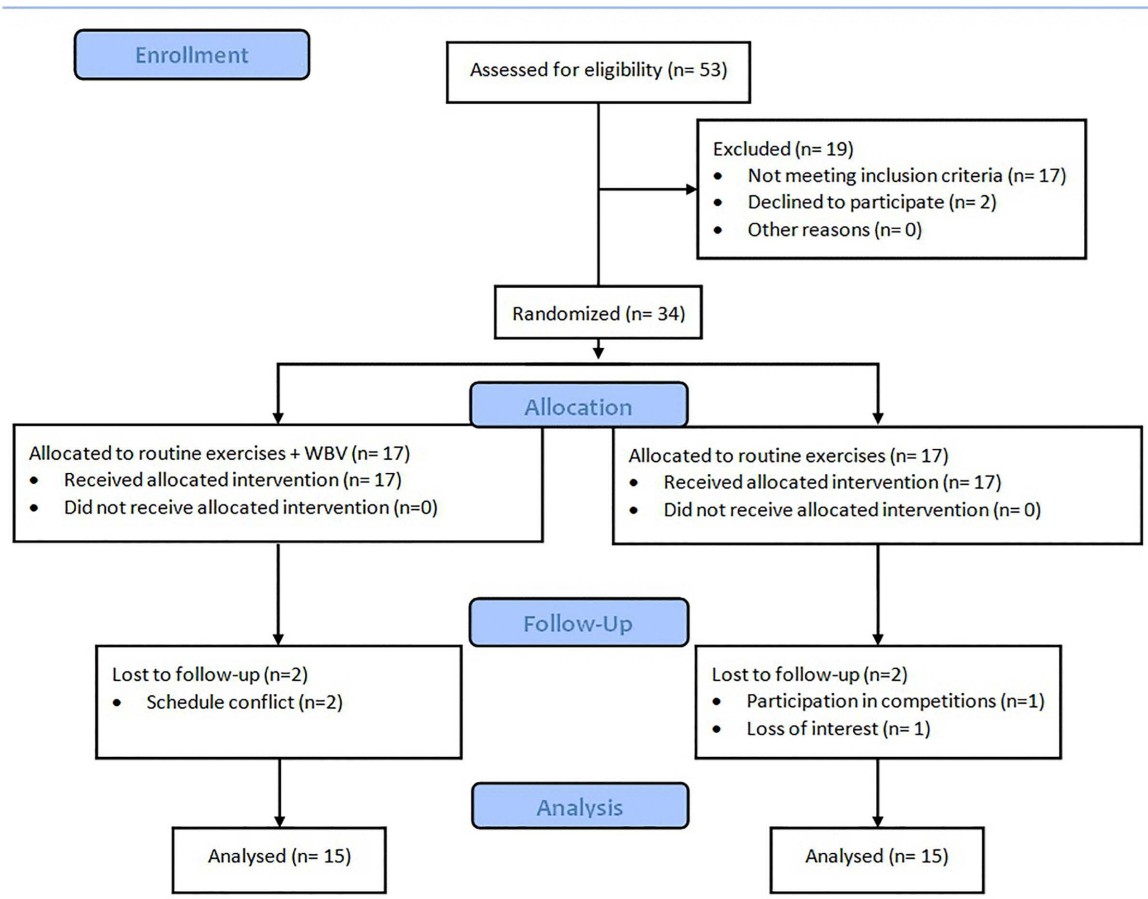

**Fig 1. The CONSORT flowchart.**

human participants were conducted in accordance with the relevant guidelines and regulations, including the ethical principles outlined in the Declaration of Helsinki. The study protocol was reviewed and approved by the Ethics Committee of Tehran University of Medical Sciences (IR.TUMS.FNM.REC.1400.211). Written informed consent was obtained from all participants prior to their enrollment in the study.

The inclusion criteria were as follows: (1) participants aged 18–40 years with a history of ACLR for over six months; (2) the operated limb must be unilateral and dominant (used to kick a ball); (3) recreationally active (engaging in sports involving jumping like volleyball, basketball, soccer, or handball at least three times a week for more than 30 minutes each session) [30]; (4) returned to sports activities with medical clearance and completed rehabilitation; (5) no pain, inflammation, or limited range of motion in the knee; (6) no contraindications to WBV (e.g., pregnancy, acute thrombosis, severe cardiovascular problems, pacemaker, discopathy, spondylosis, severe diabetes, epilepsy, acute infection, severe migraine, tumor, and kidney stones) [17]; and (7) no history of surgery or traumatic injuries to the contra-lateral limb.

The exclusion criteria included: (1) inability to perform the single-leg jump-landing task despite multiple practice attempts during the familiarization phase, indicated by consistent failure to maintain postural stability on the test limb and repeated demonstration of one or more of the following errors— involvement of the non-landing limb during takeoff or landing, additional hopping after landing, or excessive movement of the arms, trunk, or non-landing limb; (2) absence from two consecutive or three non-consecutive therapy sessions; and (3) unwillingness to continue participation.

The sample was restricted to individuals aged 18–40 years to control for age as a key confounding factor associated with declines in balance, postural stability, and neuromuscular control. With advancing age, postural sway increases, sensory integration slows, and proprioceptive as well as reflexive balance control deteriorate [31]. Such age-related neuromuscular changes may alter responsiveness to WBV, potentially affecting the magnitude and nature of postural and functional adaptations in older compared with younger individuals. By limiting the age range, we reduced the likelihood that age-related deficits would obscure or modify the observed effects of WBV on postural control and jump biomechanics.

A minimum six-month period post-ACLR was required before initiating WBV to ensure clinical safety and allow adequate healing. This timeline, based on surgical team recommendations, aimed to prevent interference with early tissue repair and ensure participants were ready for introducing additional treatments like WBV.

## 2.3. Sample size

The sample size was determined using G*Power 3.1.3 based on a pilot study of 10 participants (5 per group), focusing on the SI and TTS in the AP direction. In that pilot, the AP SI changed from 0.035 to 0.030 (–0.005) in the control group and from 0.042 to 0.036 (–0.006) in the intervention group; the AP TTS changed from 950 ms to 900 ms (–50 ms) in control and from 960 ms to 854 ms (–106 ms) in the intervention group. The calculation used a within-between interaction effect size ($\eta^2$) of 0.066 and 0.081, a power of 0.80, $\alpha = 0.05$, a non-sphericity correction ($\epsilon$) = 1, and a correlation among repeated measures of 0.50. The analysis indicated that at least 30 participants were needed to detect a within-between interaction in a 2-group, 2-measurement design with those parameters. Consequently, 34 participants were recruited to account for a potential 10% dropout rate.

## 2.4. Randomization and blinding

Participants were randomly assigned to either the intervention group (WBV plus routine exercise) or the control group (routine exercise alone) with a 1:1 allocation ratio. Random sequence generation was performed by an independent researcher using a web-based randomization service (www.randomization.com). Group allocation was concealed using sequentially numbered, opaque, sealed envelopes prepared by a third party not involved in participant recruitment, intervention delivery, or outcome assessment.

The study employed a single-blind (assessor-blinded) design. Interventions were delivered by a trained physiotherapist who was not involved in outcome assessment. Outcome measurements were performed by an independent assessor who remained blinded to group allocation to minimize detection bias. Blinding of participants was not feasible due to the perceptible mechanical stimuli produced by WBV, which cannot be convincingly replicated without an active platform, and no validated sham WBV protocol was available. Treating physiotherapists were necessarily aware of group allocation because they administered the intervention.

## 2.5. Assessments

Outcome measures assessed both before and after the treatment. Post-tests were performed within 24 to 48 hours after completion of treatment.

**2.5.1. The stability index and time to stabilizationextracted from jump-landing test.** A force plate device (Bertec Corporation, Columbus, OH, USA) was utilized with a sampling frequency of 500 Hz to test diagonal jump landings.To determine the maximum horizontal jump, participant performed three diagonal jumps. The maximum distance achieved was recorded. For safety, 75% of this maximum distance was used for jumps on the force plate. Depending on the limb tested, participants stood on both feet at either the posteromedial or posterolateral corners, at a 45-degree angle between the sagittal and frontal planes, and jumped diagonally (anteromedial or anterolateral) to land on the operated limb [7](Fig 2). Participantswere asked to regain stability as quickly as possible after landing. The jump-landing test was conducted in diagonal direction because Patterson's study showed that in individuals who underwent ligament reconstruction surgery, the TTS in the diagonal jump-landing test was significantly different compared to healthy individuals, whereas no significant difference was observed in the forward jump-landing test. Therefore, it seems that the TTSin the diagonal jump-landing test may be more sensitive in detecting abnormal landing patterns in individuals with ligament reconstruction surgery [7].

Before the main test, participants performed several practice jumps. If a participant failed to maintain balance upon landing, if the non-landing limb interfered during the jump or landing, if there was an additional small hop on landing, or if there was excessive movement in the arms, trunk, or non-landing limb causing the test limb to lift off the force plate, the jump was discarded and repeated. Data from three successful jumps, including GRFs in the x, y, and z directions, were recorded on the force plate for 10 seconds post-landing. There was a one-minute interval between each repetition [20].

The stability index in the ML and APdirections was calculated by measuring the deviation of the X and Y components of the GRF from the zero point, respectively, using equations 1 and 2. The stability index in the vertical direction was determined by measuring the deviation of the Z component of the GRF from the individual's weight (equation 3). The dynamic postural stability index combines the stability indices from the AP, ML, and vertical directions, reflecting sensitivity to changes in all three directions (equation 4). These values were normalized based on each individual's weight to allow for comparisons between participants. The average of three repetitions was used as the final data. Calculations were based on the period from the moment of landing (when the vertical component of the GRF exceeded 5% of body weight) to three seconds afterward, as this duration best mimics sports performance. Moreover, to determine body weight, participants stood on the force plate on a single limb for 5 seconds, and the average vertical GRF during this period was considered as the person's weight [20,32].

$$\text{MLSI} = \sqrt{\left( \frac{\sum (0 - GRFx)^2}{number\ of\ data\ points} \right) \div BW}$$

(1)

$$\text{APSI} = \sqrt{\left( \frac{\sum (0 - GRFy)^2}{number\ of\ data\ points} \right) \div BW}$$

(2)

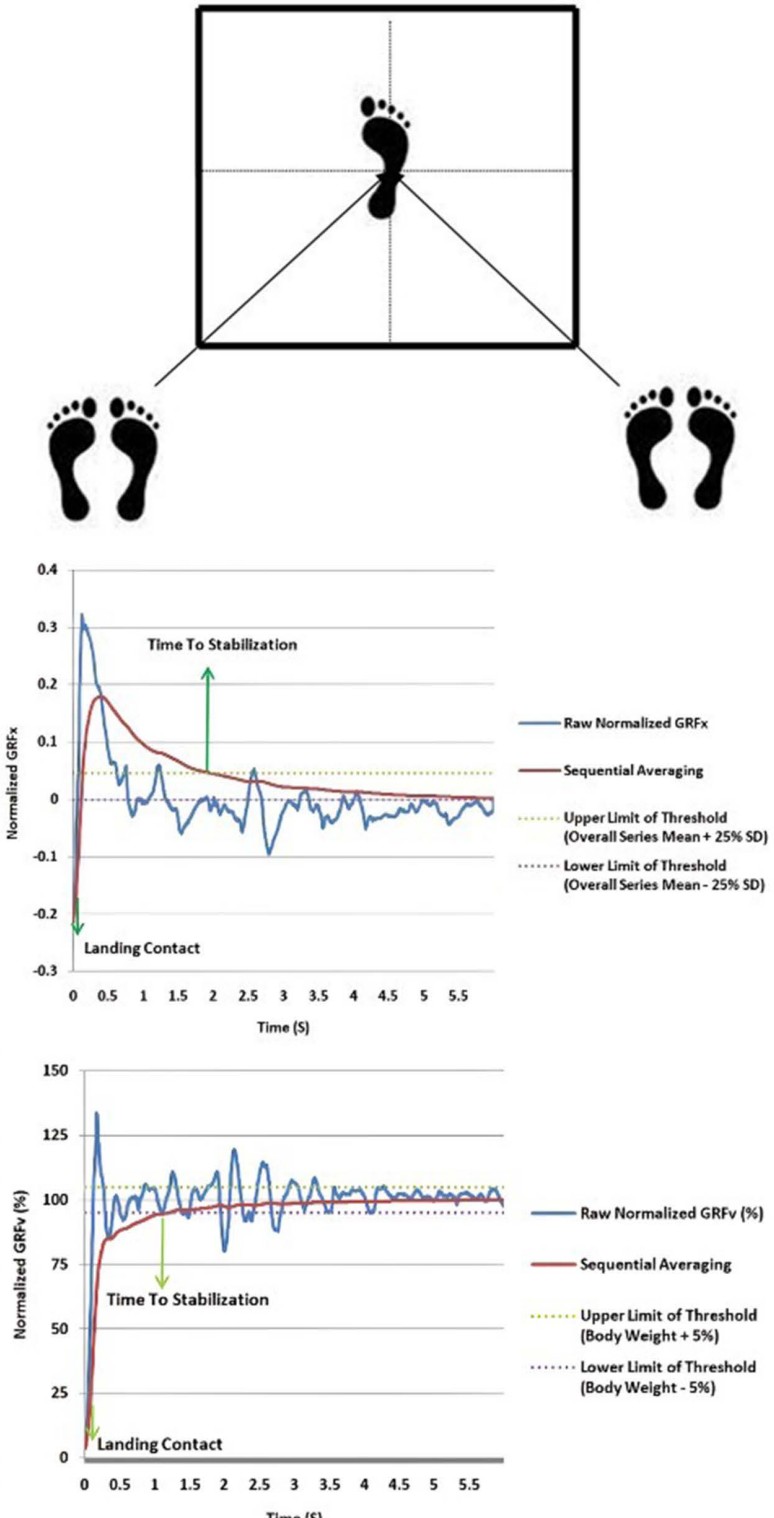

**Fig 2. From top to bottom: diagonal jump-landing test setup, including (A) force plate positioning; (B) time to stabilization (TTS) measurement in the mediolateral direction, and (C) in the vertical direction.**

$$VSI = \sqrt{\left(\frac{\sum (BW - GRFz)^2}{number\ of\ data\ points}\right) \div BW} \tag{3}$$

$$DPSI = \sqrt{\left(\frac{\sum (0 - GRFx)^2 + \sum (0 - GRFy)^2 + \sum (BW - GRFz)^2}{number\ of\ data\ points}\right) \div BW} \tag{4}$$

To calculate time to stabilization, a time series was generated by sequentially averaging the GRFs—normalized to each participant's body weight—across the AP, ML, and vertical directions. This involved cumulatively adding each successive GRF data point to the previous one and computing the average, thereby producing a continuous time series that reflects post-landing stabilization. Normalizing GRFs to body weight accounts for inter-individual differences in mass, enabling meaningful comparisons across participants. Additionally, the standard deviation and average time series of the normalized GRF in the AP and ML directions during the first 3 seconds after landing were calculated. A threshold was set as "standard deviation ± 0.25 of the mean." Stability was indicated when the time series from sequential averaging fell within this threshold range. The time point at which the time series entered this range was considered the moment of stability, and the interval between this moment and the landing moment was defined as the TTS (Fig 2). For the vertical GRF, the threshold was "±5% of the person's weight". The RVTTS was also calculated using equation 5, and the average of three repetitions was considered the final data for each direction [20,32].

$$RVTTS = \sqrt{(MLTTS)^2 + (APTTS)^2} \tag{5}$$

**2.5.2 Y-Balance Test (YBT).** The Y-balance test was used to assess dynamic balance. In this test, participants stood on the operated limb and moved the other limb along three specified directions on the ground: anterior, PM, and PL. The PM and PL directions were positioned at 90-degree angles relative to each other, with both forming 135-degree angles with the anterior direction.Participantswere instructed to touch the farthest possible point along each direction with their moving foot while maintaining balance. The distance from the center to the touched point was measured. To normalize the results, the distance was divided by the length of the lower limb (measured from the anterior superior iliac spine to the center of the medial malleolus) and then multiplied by 100 to obtain a percentage value relative to the lower limb length. Each participant completed the test three times in each direction, and the average distance from the three trials was utilized for data analysis.If the participant's stepping foot touched any point other than the intended endpoint, if the supporting foot lifted off the ground, or if the participant lost balance, the trial was considered invalid and repeated. Previous studies have demonstrated that the YBT is a reliable and valid tool for measuring dynamic balance [33].

**2.5.3 Limb Symmetry Index (LSI) in the 6-Meter Timed Hop (6MTH) Test.** The 6m-timed hop test was utilized to evaluate the limb symmetry index. During this test, participants were instructed to hop on one leg over a defined distance of 6 metersas quickly as possible. The time was measured using a standard stopwatch, starting when the participant's heel lifted off the ground and ending upon crossing the finish line. The test was conducted for both limbs, and the duration taken for the healthy limb was divided by the duration for the operated limb, and then multiplied by 100. A higher LSI indicates better physical performance [34].

## 2.6 Groups

**2.6.1 Control group.** In this group, routine exercises were performed, which included strength and perturbation-based balance training. The treatment period consisted of 12 sessions (4 weeks, with 3 sessions per week). The exercise procedures are detailed in Table 2 [16,35,36].

**Table 2. The protocol of routine exercises.**

| Session | Strength Training | | | | | Perturbation-Based Balance Training (each exercise consists of 2 sets of 2 repetitions, with a 30-second rest time between sets) | |
|---|---|---|---|---|---|---|---|
| | Muscle Groups | | | Intensity | Sets | Repetitions | Rest Time Between Sets |
| 1 | Hip abductors & adductors | 10RM | 3 | 8 | 60 s | • Bilateral stance on a rocker board<br>• Perturbation in the AP direction | • The injured limb on a roller board and the healthy limb on a platform<br>• Perturbation in the AP direction<br>• Exchanging the injured limb with the healthy one |
| 2 | Hip abductors & adductors + external rotators | 10RM | 3 | 8 | 60 s | • Bilateral stance on a rocker board<br>• Perturbation in the ML direction | • The injured limb on a roller board and the healthy limb on a platform<br>• Perturbation in the ML direction<br>• Exchanging the injured limb with the healthy one |
| 3 | Hip abductors & adductors & external rotators + extensors | 10RM | 3 | 8 | 60 s | • Unilateral stance on a rocker board<br>• Perturbation in the AP direction<br>• Perturbation in the ML direction | • The injured limb on a roller board and the healthy limb on a platform<br>• Perturbation in the AP direction<br>• Perturbation in the ML direction<br>• Exchanging the injured limb with the healthy one |
| 4 | Hip abductors & adductors & external rotators & extensors + leg press | 10RM | 4 *<br>3** | 8 | 50 s | • Unilateral stance on a rocker board<br>• Perturbation in the AP direction<br>• Perturbation in the ML direction<br>• Perturbation in diagonal direction | • The injured limb on a roller board and the healthy limb on a platform<br>• Perturbation in the AP direction<br>• Perturbation in the ML direction<br>• Perturbation in the rotation<br>• Exchanging the injured limb with the healthy one |
| 5 | Hip abductors & adductors & external rotators & extensors & leg press + leg curl | 10RM | 4 *<br>3** | 10 | 50 s | • Unilateral stance on a rocker board<br>• Perturbation in the AP direction<br>• Perturbation in the ML direction<br>• Perturbation in diagonal direction | • The injured limb on a roller board and the healthy limb on a platform<br>• Perturbation in the AP direction<br>• Perturbation in the ML direction<br>• Perturbation in the rotation<br>• Exchanging the injured limb with the healthy one |
| 6 | Hip abductors & adductors & external rotators & extensors & leg press & leg curl + quadriceps | 10RM | 4 *<br>3** | 10 | 40 s | • Unilateral stance on a rocker board<br>• Perturbation in the AP direction<br>• Perturbation in the ML direction<br>• Perturbation in diagonal direction | • The injured limb on a roller board and the healthy limb on a platform<br>• Perturbation in the AP direction<br>• Perturbation in the ML direction<br>• Perturbation in the rotation<br>• Exchanging the injured limb with the healthy one |
| 7 | Hip abductors & adductors & external rotators & extensors & leg press & leg curl + quadriceps | 10RM | 4 | 10 | 40 s | • Unilateral stance on a rocker board<br>• Perturbation in the AP direction<br>• Perturbation in the ML direction<br>• Perturbation in diagonal direction<br>• Throwing ball against wall | • The injured limb on a roller board and the healthy limb on a platform<br>• Perturbation in the AP direction<br>• Perturbation in the ML direction<br>• Perturbation in the rotation<br>• Throwing ball against wall<br>• Exchanging the injured limb with the healthy one |
| 8 | Hip abductors & adductors & external rotators & extensors & leg press & leg curl & quadriceps + mini squat | 10RM 1/10 BW | 4 *<br>3** | 10 | 40 s | • Unilateral stance on a rocker board<br>• Perturbation in the AP direction<br>• Perturbation in the ML direction<br>• Perturbation in diagonal direction<br>• Throwing ball against wall/floor | • The injured limb on a roller board and the healthy limb on a platform<br>• Perturbation in the AP direction<br>• Perturbation in the ML direction<br>• Perturbation in the rotation<br>• Throwing ball against wall/floor<br>• Exchanging the injured limb with the healthy one |

*(Continued)*

| Session | Strength Training | | | | | Perturbation-Based Balance Training (each exercise consists of 2 sets of 2 repetitions, with a 30-second rest time between sets) | |
|---|---|---|---|---|---|---|---|
| | **Muscle Groups** | | | **Intensity** | **Sets** | **Repetitions** | **Rest Time Between Sets** |
| 9 | Hip abductors & adductors & external rotators & extensors & leg press & leg curl & quadriceps + mini squat | 10RM 1/10 BW | 4 | 10 | 40 s | • Unilateral stance on a rocker board<br>• Perturbation in the AP direction<br>• Perturbation in the ML direction<br>• Perturbation in diagonal direction<br>• Throwing ball against wall/floor<br>• Thrown by other | • The injured limb on a roller board and the healthy limb on a platform<br>• Perturbation in the AP direction<br>• Perturbation in the ML direction<br>• Perturbation in the rotation<br>• Throwing ball against wall/floor<br>• Thrown by other<br>• Exchanging the injured limb with the healthy one |
| 10 | Hip abductors & adductors & external rotators & extensors & leg press & leg curl & quadriceps + squat | 10RM 1/8 BW | 4 * 3** | 10 | 30 s | • Unilateral stance on a rocker board<br>• Perturbation in the AP direction<br>• Perturbation in the ML direction<br>• Perturbation in diagonal direction<br>• Throwing ball against wall/floor<br>• Thrown by other<br>• Other individually adjusted relevant sport-specific activities | • The injured limb on a roller board and the healthy limb on a platform<br>• Perturbation in the AP direction<br>• Perturbation in the ML direction<br>• Perturbation in the rotation<br>• Throwing ball against wall/floor<br>• Thrown by other<br>• Other individually adjusted relevant sport-specific activities<br>• Exchanging the injured limb with the healthy one |
| 11 | Hip abductors & adductors & external rotators & extensors & leg press & leg curl & quadriceps + squat | 10RM 1/8 BW | 4 | 10 | 30 s | • Unilateral stance on a rocker board<br>• Perturbation in the AP direction<br>• Perturbation in the ML direction<br>• Perturbation in diagonal direction<br>• Throwing ball against wall/floor<br>• Thrown by other<br>• Other individually adjusted relevant sport-specific activities | • The injured limb on a roller board and the healthy limb on a platform<br>• Perturbation in the AP direction<br>• Perturbation in the ML direction<br>• Perturbation in the rotation<br>• Throwing ball against wall/floor<br>• Thrown by other<br>• Other individually adjusted relevant sport-specific activities<br>• Exchanging the injured limb with the healthy one |
| 12 | Hip abductors & adductors & external rotators & extensors & leg press & leg curl & quadriceps + squat | 10RM 1/6 BW | 4 | 10 | 30 s | • Unilateral stance on a rocker board<br>• Perturbation in the AP direction<br>• Perturbation in the ML direction<br>• Perturbation in diagonal direction<br>• Throwing ball against wall/floor<br>• Thrown by other<br>• Other individually adjusted relevant sport-specific activities | • The injured limb on a roller board and the healthy limb on a platform<br>• Perturbation in the AP direction<br>• Perturbation in the ML direction<br>• Perturbation in the rotation<br>• Throwing ball against wall/floor<br>• Thrown by other<br>• Other individually adjusted relevant sport-specific activities<br>• Exchanging the injured limb with the healthy one |

10RM: 10 repetition maximum. 10RM is the maximum weight a person can lift for exactly 10 repetitions of a specific exercise.

10RM was changed every week.

BW: body weight.

* Number of sets for muscle groups already trained.

** Number of sets for muscle groups that are newly added.

The design of the exercise protocol was grounded in existing literature and clinical reasoning, focusing on two critical aspects in post-ACL rehabilitation: neuromuscular control and muscle strength. These components are essential for optimizing recovery and minimizing re-injury risk. The perturbation-based elements were inspired by the protocol of Eitzen et al. (2010), who demonstrated significant improvements in knee function and dynamic stability through a structured, progressive program incorporating neuromuscular and balance training [35]. This evidence supports the use of perturbation training to enhance proprioceptive feedback and improve functional stability, both of which are key in ACL rehabilitation.

In parallel, the strengthening component was adapted from Moezy et al. (2008), whose findings underscored the effectiveness of conventional resistance training in improving postural control in athletes following ACLR [16]. This body of work highlights the critical role of strengthening exercises in restoring muscle strength and stability, which are often compromised after ACL surgery.

It is also important to note that both the studies by Eitzen et al. (2010) and Moezy et al. (2008) have reported no significant adverse effects associated with these exercises, supporting their safety for use in post-ACL rehabilitation programs. Therefore, we can be confident in their safe integration into rehabilitation programs without causing additional harm to participants.

By integrating these two complementary approaches—perturbation-based neuromuscular training and resistance-based strengthening exercises—our protocol offers a comprehensive, evidence-based framework that addresses both the functional stability and strength deficits commonly observed in ACLR athletes.

The progression of exercise intensity was guided by established principles of progressive overload, task specificity, and neuromuscular adaptation. This graded progression was designed to continuously challenge the sensorimotor system while ensuring participant safety, allowing sufficient time for physiological adaptation, and minimizing the risk of excessive fatigue or overuse injury. The progression strategy was grounded in prior published evidence, particularly the protocols described by Eitzen et al. (2010) and Moezy et al. (2008), and was implemented under strict safety monitoring [16,35]. All training sessions were conducted under close clinical supervision, and participants were continuously monitored for pain, discomfort, or abnormal fatigue. No adverse events, neuromuscular fatigue, or overuse injuries were reported during the exercise.

The initial training intensity was determined using a standardized 10-repetition maximum (10RM) testing protocol [37]. At first, participants performed a standardized warm-up consisting of 5–10 repetitions at approximately 40–50% of their perceived maximal effort for the target muscle group, followed by a 1-minute rest period before the main testing trial. The resistance was then progressively increased in small increments of 5–10%. During each trial, participants performed up to 10 repetitions with correct technique. The load was adjusted in subsequent trials—if more than 10 repetitions were completed, the weight was increased, whereas if fewer than 10 repetitions were achieved, the weight was decreased after adequate rest—until the maximum load that could be lifted for exactly 10 repetitions with proper form and full range of motion was identified as the individual 10RM. Rest intervals of 2–3 minutes were provided between trials to minimize fatigue, and testing of different muscle groups was separated by sufficient recovery periods to prevent cumulative fatigue [37].

**2.6.2 Intervention group.** In this group, WBV using the Power Plate® Next Generation (Power Plate North America, Northbrook, IL, USA) was incorporated alongside routine exercises. Participants in each session first received WBV and then performed the exercises. The treatment period consisted of 12 sessions (4 weeks, with 3 sessions per week). The WBV protocol is detailed in Table 3 and illustrated in Fig 3 [16,36]. To ensure standardization of positions P1–P9 during the WBV intervention, all exercises were performed under the supervision of a trained physiotherapist throughout the intervention sessions. Clear verbal instructions and consistent cues were given for each position, with attention to body alignment, joint angles (e.g., slight or 90-degree knee flexion), and posture (e.g., straight back). The vibration platform was marked to guide foot placement and ensure uniform exposure across participants. These procedures helped minimize variability and ensured consistent.

**Table 3. The protocol of whole body vibration.**

| Session | Frequency | Amplitude | Number of Sets | Duration of Each Set | Rest Time Between Sets | Duration of Vibration application | Different Positions of Standing on the Plate and Number of Sets for Each | | | | | | | | |
|---|---|---|---|---|---|---|---|---|---|---|---|---|---|---|---|
| | | | | | | | P1 | P2 | P3 | P4 | P5 | P6 | P7 | P8 | P9 |
| 1 | 30 Hz | Low | 8 | 30 s | 60 s | 4 min | 2 | 2 | 1 | – | 1 | – | – | 1 | 1 |
| 2 | 30 Hz | Low | 11 | 30 s | 60 s | 5.5 min | 3 | 3 | 2 | – | 1 | – | – | 1 | 1 |
| 3 | 30 Hz | Low | 13 | 30 s | 60 s | 6.5 min | 3 | 3 | 3 | – | 1 | – | – | 1 | 2 |
| 4 | 35 Hz | Low | 16 | 30 s | 60 s | 8 min | 3 | 3 | 3 | 1 | 2 | – | 1 | 1 | 2 |
| 5 | 35 Hz | Low | 16 | 45 s | 60 s | 12 min | 2 | 2 | 3 | 2 | 2 | – | 1 | 2 | 2 |
| 6 | 35 Hz | Low | 16 | 45 s | 60 s | 12 min | 2 | 2 | 3 | 2 | 2 | – | 1 | 2 | 2 |
| 7 | 40 Hz | High | 18 | 45 s | 60 s | 13.5 min | 2 | 2 | 3 | 2 | 2 | 1 | 2 | 2 | 2 |
| 8 | 40 Hz | High | 20 | 45 s | 60 s | 15 min | 2 | 2 | 3 | 2 | 3 | 1 | 2 | 2 | 3 |
| 9 | 40 Hz | High | 20 | 45 s | 60 s | 15 min | 2 | 2 | 3 | 2 | 3 | 1 | 2 | 2 | 3 |
| 10 | 50 Hz | High | 16 | 60 s | 60 s | 16 min | 1 | 1 | 2 | 2 | 2 | 2 | 2 | 2 | 2 |
| 11 | 50 Hz | High | 16 | 60 s | 60 s | 16 min | 1 | 1 | 2 | 2 | 2 | 2 | 2 | 2 | 2 |
| 12 | 50 Hz | High | 16 | 60 s | 60 s | 16 min | 1 | 1 | 2 | 2 | 2 | 2 | 2 | 2 | 2 |

P1: Limbs positioned in the middle of the plate, slightly apart, knees slightly bent and back straight.

P2: Standing on one limb with the foot in the middle of the plate, knee slightly bent and back straight.

P3: Mini squat.

P4: Mini squat on one limb.

P5: Squat.

P6: Squat on one limb.

P7: Squat with limbs apart.

P8: One limb in the middle of the plate and the other limb outside the plate, knees bent at 90 degrees, and back straight.

P9: Standing on the toes.

The choice to administer WBV prior to exercises was guided by its potential neuromodulatory influence, which may enhance motor cortex excitability and improve subsequent neuromuscular performance [30,36]. WBV repeatedly introduces sensory input to the central nervous system (CNS) by activating muscle spindles and generating afferent signals toward the spinal cord and brain. This afferent barrage may engage neural circuits at both spinal and supra-spinal levels, thereby increasing cortico-spinal excitability and supporting enhanced motor output during the exercises that follow [30,36].

Evidence from previous research reinforces this sequencing approach. Pamukoff et al. and Khanmohammadi et al. have shown that WBV can boost motor cortex excitability in individuals after ACLR [30,36]. Similarly, studies by Krause et al. and Mileva et al. in healthy participants indicate that WBV can acutely enhance cortico-spinal excitability and reduce spinal inhibition, pointing to a modulatory effect on neural activity within the primary motor cortex and descending pathways [38,39].Taken together, these neurophysiological responses suggest that applying WBV before exercise may prime the neuromuscular system, enabling more effective muscle activation and improved muscular performance during subsequent training tasks.

The vibration frequency, number of platform positions, and total exposure time were progressively increased across sessions in accordance with the protocol described by Moezy et al. (2008) [16]. This stepwise progression was designed to promote gradual neuromuscular adaptation while minimizing the risk of excessive fatigue or overuse injury. All sessions were conducted under close clinical supervision, with continuous monitoring for pain, discomfort, or abnormal fatigue. No adverse events were reported throughout the intervention period.

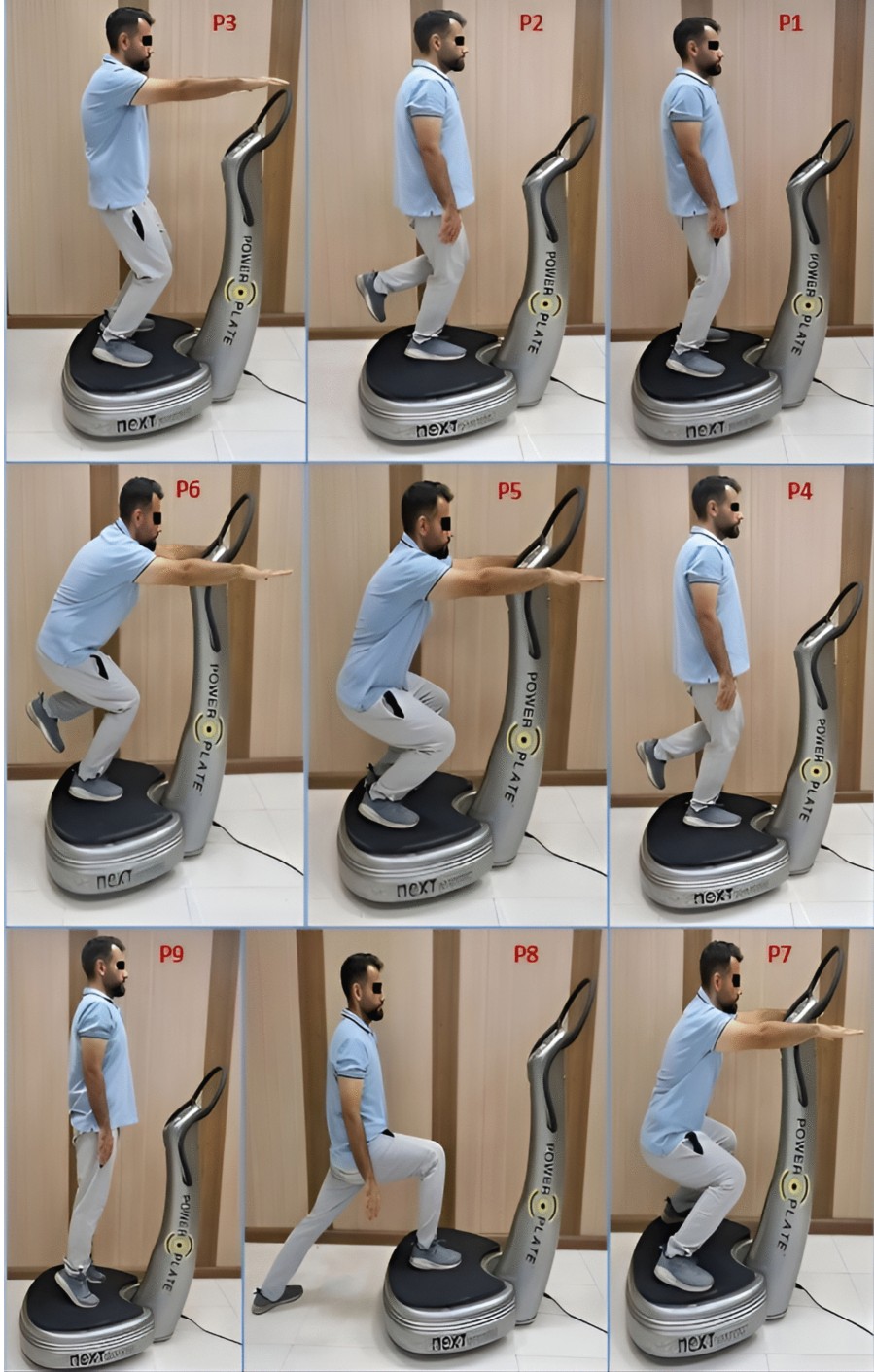

**Fig 3. The protocol of whole body vibration. P1:** Limbs positioned in the middle of the plate, slightly apart, knees slightly bent and back straight; **P2:** Standing on one limb with the foot in the middle of the plate, knee slightly bent and back straight; **P3:** Mini squat; **P4:** Mini squat on one limb; **P5:** Squat; **P6:** Squat on one limb; **P7:** Squat with limbs apart; **P8:** One limb in the middle of the plate and the other limb outside the plate, knees bent at 90 degrees, and back straight; **P9:** Standing on the toes.

## 2.7 Statistical analysis

The Shapiro-Wilk test was employed to assess the normality of the data distribution, which was found to be normally distributed. The independent t-test and chi-square test were performed for quantitative and qualitative data, respectively, to compare the two groups at baseline. The results indicated no significant differences between the groups, showing they were comparable at baseline. A two-way mixed ANOVA with one within-subjects factor and one between-groups factor was conducted to examine the main effects of group (routine exercises and routine exercises plus WBV), time (before and after) and their interactions. Prior to conducting the mixed ANOVA, the assumption of homogeneity of variances was assessed using Levene's test, which indicated that variances were comparable between groups. Because the within-subject factor (time) included only two levels (before and after), the sphericity assumption was inherently met and did not require formal testing. When the interactions were significant, separate one-way repeated measures ANOVAs were performed for each group to delve deeper into these interactions. Additionally, partial eta squared ($\eta$2) was reported as a measure of effect size, with small (0.01–0.06), moderate (0.06–0.14), and large ($\geq$ 0.14) categories.

## 3. Results

The descriptive data and the results of ANOVA are presented in Tables 4 and 5, respectively.

### 3.1. TTS

The interaction between group and time for TTS in the AP direction was significant ($F_{(1,28)}$= 14.529, P = 0.001, $\eta^2$ = 0.342), indicating that the groups exhibited different changes over time. In the intervention group, the time effect was significant ($F_{(1,14)}$= 18.409, P = 0.001, $\eta^2$ = 0.568), whereas no significant time effect was found in the control group ($F_{(1,14)}$= 2.478, P = 0.138, $\eta^2$ = 0.150). Specifically, the intervention group, which received WBV in addition to routine exercises, showed a significant reduction in TTS after treatment, with a large effect size. The main effect of group was not significant ($F_{(1,28)}$= 1.882, P = 0.181, $\eta^2$ = 0.063)[Tables 4 and 5].

**Table 4. The mean and standard deviation of parameters.**

| Parameters | Pre-Test | | | | Post-Test | | | |
|---|---|---|---|---|---|---|---|---|
| | Intervention | | Control | | Intervention | | Control | |
| | Mean | SD | Mean | SD | Mean | SD | Mean | SD |
| **Jump-Landing Test** | | | | | | | | |
| AP TTS (ms) | 959.78 | 446.66 | 940.82 | 288.71 | 697.58 | 458.71 | 1071.47 | 297.25 |
| ML TTS (ms) | 1881.60 | 96.32 | 1890.88 | 68.38 | 1870.27 | 70.35 | 1895.56 | 51.19 |
| V TTS (ms) | 1778.80 | 860.69 | 1709.08 | 1084.71 | 1793.41 | 715.36 | 1676.22 | 690.40 |
| RV TTS (ms) | 2157.67 | 238.97 | 2149.16 | 162.74 | 2028.68 | 249.03 | 2210.83 | 142.99 |
| AP SI | 0.04 | 0.02 | 0.03 | 0.01 | 0.04 | 0.01 | 0.03 | 0.01 |
| ML SI | 0.15 | 0.04 | 0.13 | 0.03 | 0.14 | 0.02 | 0.14 | 0.02 |
| V SI | 0.21 | 0.07 | 0.19 | 0.03 | 0.21 | 0.06 | 0.19 | 0.03 |
| DPSI | 0.24 | 0.06 | 0.25 | 0.03 | 0.24 | 0.06 | 0.25 | 0.03 |
| **Clinical Tests** | | | | | | | | |
| A YBT (%) | 93.05 | 13.54 | 96.86 | 21.65 | 99.64 | 16.41 | 101.65 | 25.20 |
| PM YBT (%) | 92.90 | 12.06 | 96.72 | 13.52 | 100.14 | 12.50 | 102.18 | 15.39 |
| PL YBT (%) | 87.02 | 9.41 | 88.88 | 9.70 | 96.29 | 9.19 | 97.21 | 13.86 |
| LSI in 6MTH (%) | 0.75 | 0.12 | 0.73 | 0.17 | 0.86 | 0.08 | 0.82 | 0.13 |

AP = Anteroposterior; ML = Mediolateral; V = Vertical; RV = Resultant Vector; TTS = Time to Stabilization; SI = Stability Index; DPSI = Dynamic Postural Stability Index; YBT = Y Balance Test; A = Anterior; PM = Posteromedial; PL = Posterolateral; LSI in 6MTH = Limb Symmetry Index in the 6-Meter Timed Hop Test; SD = Standard Deviation.

**Table 5. The ANOVA results.**

| Parameters | Time Effect | | | Group Effect | | | Time* Group Effect | | |
|---|---|---|---|---|---|---|---|---|---|
| | F | P | η2 | F | P | η2 | F | P | η2 |
| **Jump-Landing Test** | | | | | | | | | |
| AP TTS | 1.629 | 0.212 | 0.055 | 1.882 | 0.181 | 0.063 | 14.529 | **0.001*** | 0.342 |
| ML TTS | 0.103 | 0.750 | 0.004 | 0.490 | 0.490 | 0.017 | 0.598 | 0.446 | 0.021 |
| V TTS | 0.003 | 0.957 | 0.000 | 0.126 | 0.725 | 0.004 | 0.020 | 0.887 | 0.001 |
| RV TTS | 1.720 | 0.200 | 0.058 | 1.546 | 0.224 | 0.052 | 13.793 | **0.001*** | 0.330 |
| AP SI | 0.720 | 0.403 | 0.025 | 2.495 | 0.125 | 0.082 | 0.819 | 0.373 | 0.028 |
| ML SI | 0.041 | 0.841 | 0.001 | 0.323 | 0.574 | 0.011 | 1.872 | 0.182 | 0.063 |
| V SI | 0.024 | 0.878 | 0.001 | 2.488 | 0.126 | 0.082 | 0.004 | 0.950 | 0.000 |
| DPSI | 0.430 | 0.517 | 0.015 | 0.024 | 0.878 | 0.001 | 0.197 | 0.660 | 0.007 |
| **Clinical Tests** | | | | | | | | | |
| A YBT | 13.204 | **0.001*** | 0.320 | 0.171 | 0.682 | 0.006 | 0.329 | 0.571 | 0.012 |
| PM YBT | 6.625 | **0.016*** | 0.191 | 0.479 | 0.495 | 0.017 | 0.130 | 0.721 | 0.005 |
| PL YBT | 33.076 | **<0.001*** | 0.542 | 0.148 | 0.703 | 0.005 | 0.096 | 0.759 | 0.003 |
| LSI in 6MTH | 43.730 | **<0.001*** | 0.610 | 0.568 | 0.457 | 0.020 | 0.528 | 0.474 | 0.019 |

AP = Anteroposterior; ML = Mediolateral; V = Vertical; RV = Resultant Vector; TTS = Time to Stabilization; SI = Stability Index; DPSI = Dynamic Postural Stability Index; YBT = Y Balance Test; A = Anterior; PM = Posteromedial; PL = Posterolateral; LSI in 6MTH = Limb Symmetry Index in the 6-Meter Timed Hop Test.

η2 is effect size (small = 0.01–0.06, medium = 0.06–0.14 and large≥ 0.14).

* Significant differences (P<0.05).

For the resultant TTS, the interaction effect of group and time was significant ($F_{(1,28)}$= 13.793, P = 0.001, η2 = 0.330). In the intervention group, the effect of time was significant ($F_{(1,14)}$=15.991, P = 0.001, η2 = 0.533), whereas in the control group, it was not significant ($F_{(1,14)}$=2.385, P = 0.145, η2 = 0.146). Specifically, in the intervention group, which received WBV in addition to routine exercises, TTS significantly decreased from pre- to post-treatment, with a large effect size. The main effect of group was not significant ($F_{(1,28)}$= 1.546, P = 0.224, η2 = 0.052)[Tables 4 and 5].

For TTS in the ML and vertical directions, no significant main effects of time, group, or time×group interaction were observed (F ≤ 0.598, p ≥ 0.446, η² ≤ 0.021 for all comparisons)[Tables 4 and 5].

### 3.2. SI

No significant main effects of time, group, or time×group interaction were found for any of the stability index components, including AP SI, ML SI, V SI, and DPSI (F ≤ 2.495, p ≥ 0.125, η² ≤ 0.082 for all comparisons). Detailed statistical results can be found in Tables 4 and 5.

### 3.3. YBT

For the YBT in all directions, a significant main effect of time was found (F ≤ 33.076, p ≤ 0.016, η² ≤ 0.542), while neither the main effect of group nor the time×group interaction reached significance (F ≤ 0.479, p ≥ 0.495, η² ≤ 0.017). These results suggest similar improvements over time in both groups, with no evidence of group-related differences. Effect size analysis indicated substantial pre-to-post treatment gains in each group)[Tables 4 and 5].

### 3.4. LSI

For the LSI, no significant interaction effect was observed ($F_{(1,28)}$ = 0.528, p = 0.474, η² = 0.019), indicating that the groups responded similarly over time. A significant main effect of time was found ($F_{(1,28)}$ = 43.730, p < 0.001, η² = 0.610),

reflecting a notable improvement in LSI on the hop test from pre- to post-intervention across both groups. The main effect of group was not significant (F(1,28) = 0.568, p = 0.457, η² = 0.020), suggesting no difference between groups. Effect size analysis confirmed that the magnitude of change following treatment was large in both groups)[Tables 4 and 5].

## 4. Discussion

The study hypothesized that integrating WBV with routine exercises would enhance dynamic postural stability during jump-landing—reflected by a greater reduction in the SI and shorter TTS—as well as lead to superior improvements in clinical assessments, compared to routine exercises alone. The results partially supported this hypothesis with respect to dynamic postural stability. Specifically, the addition of WBV led to significantly shorter TTS in the AP and resultant directions, indicating faster postural recovery following landing. However, no significant improvements were observed in the SI. In contrast, the second part of the hypothesis—regarding superior improvements in clinical assessments—was not supported. Both groups improved in the YBT and LSI in the hop test, but no significant between-group differences were found. These findings indicate that exercise therapy alone leads to improvements in general clinical measures, while the addition of WBV appears to provide limited, TTS-specific enhancement during the jump-landing task.

Although neuromuscular parameters were not directly assessed in this study, the observed reductions in AP and resultant TTS may be partially influenced by underlying neuromuscular factors. These could theoretically include changes in lower-limb muscle activation patterns, inter-muscular coordination between the quadriceps and hamstrings, or improvements in muscle strength, all of which have been suggested as important for maintaining anterior-posterior joint stability following ACL injury [40]. Furthermore, emerging evidence indicates that fatigue often induces similar neuromuscular and biomechanical responses in ACLR patients and healthy individuals; however, it can also reveal ACL-specific vulnerabilities during landing tasks [13]. In this context, the observed improvements in TTS may suggest enhanced resilience against fatigue-related biomechanical alterations, potentially supporting more stable performance under load. However, these interpretations remain speculative in the absence of direct measurements.

WBV has been shown to modulate muscle activation patterns, including alternating concentric and eccentric contractions that may enhance coordination between agonist and antagonist muscles [41,42]. These neuromuscular adaptations could theoretically facilitate faster postural responses and improved stabilization during dynamic tasks. However, no direct electromyographic (EMG) recordings, strength, or neural measurements were collected in the present study, and the link between these adaptations and reductions in TTS remains speculative.

Proper muscle activation patterns are crucial for maintaining dynamic postural control. Nevertheless, even after returning to play, individuals post-ACLR often exhibit altered activation patterns during functional tasks such as running and jump-landing. These alterations typically include reduced quadriceps activity, increased hamstring activity, and modified activation of the hip abductors and adductors [43,44]. These altered muscle activation patterns may serve as a protective mechanism against re-injury; however, they can also limit the lower limb's capacity for rapid and efficient movements, potentially contributing to prolonged TTS during jump-landing [45]. WBV has been shown to modulate muscle activity, including inhibitory effects on medial hamstrings during knee flexion in healthy individuals, suggesting its potential to restore a more balanced activation pattern [45].

Quadriceps inhibition after ACL injury, known as arthrogenic inhibition, arises from both spinal and supra-spinal mechanisms. At the spinal level, joint injury, swelling, and pain disrupt the gamma motor neuron loop, increase Ib interneuron inhibition, and enhance the flexion reflex, collectively reducing quadriceps activity [46]. Supra-spinal centers, which process proprioceptive inputs, also show decreased excitability, leading to reduced motor output and lower motor cortex activity [47]. Evidence indicates that WBV can mitigate muscle inhibition, enhance motor cortex excitability, and increase cortico-spinal pathway excitability in both post-ACL and healthy individuals [30,36,38,48,49]. Consequently, WBV may enhance neural drive to knee-stabilizing muscles, which could contribute to improved muscle performance and dynamic stability, potentially reflected in changes in jump-landing kinetics.

The observed improvement in dynamic postural stability during jump-landing following the addition of WBV to routine exercises may be partially explained by enhanced proprioceptive function, which could lead to better awareness of joint position and movement velocity in the lower limbs. Supporting this interpretation, Moezy et al. (2008) demonstrated that WBV improved knee joint proprioception in individuals following ACLR [16]. In addition, two recent systematic reviews (2023) in other musculoskeletal conditions, including low back pain and ankle instability, reported that WBV interventions are associated with significant improvements in joint proprioception [50,51]. However, it should be noted that proprioception was not directly assessed in the present study.

To our knowledge no study has specifically investigated the effect of WBV on dynamic postural stability during jump-landing in individuals who have undergone ACLR. While some research has assessed the effectiveness of WBV on postural stability in this population, these studies have typically employed simple tests where participants stood on a Biodex or force plate and tried to maintain their COG within the base of support [15–19].These studies generally demonstrate a positive impact of WBV on balance. However, among the five studies conducted, one—which examined the immediate, single-session effects of WBV—did not report significant improvements in balance [19].

Although these studies demonstrated positive effects of WBV, they vary in terms of postural stability evaluation methods, control groups, treatment durations, and participant characteristics. In the 2014 study by Berschin et al., one group received WBV, while the other group underwent a standard rehabilitation program for nine weeks, starting two weeks after surgery. Vibration frequency increased from 10 to 30 Hz, with tasks progressing from isometric standing on the WBV platform to dynamic exercises like squats. Postural stability was assessed using the Biodex stability system [15]. In 2013, Fu et al. conducted a study where one group received routine rehabilitation, and the other group received routine rehabilitation along with vibration therapy for eight weeks, starting one month after surgery. The WBV group underwent 16 treatment sessions with increasing vibration frequencies of 35–50 Hz and constant amplitude of 4 mm, with a gradual increase in the exercise-to-rest ratio. Postural stability was assessed using the Biodex stability system [17]. In 2008, Moezy et al. compared WBV with traditional exercises, including proprioception and flexibility exercises along with progressive strengthening exercises over four weeks, starting three months post-surgery. Over 12 sessions, vibration frequency increased from 30 to 50 Hz, amplitude from 2.5 to 5 mm, and duration from 4 to 16 minutes. Postural stability was assessed using the Biodex stability system [16]. Another study by Pistone et al. in 2016 investigated the effects of 12 sessions of WBV in addition to a traditional rehabilitation program, compared to a traditional rehabilitation program alone, starting one month after surgery.In this study, postural sway was measured during quiet standing by force plate [18].

Both the current study and the previously mentioned studies investigated the effects of multiple WBV sessions. However, a 2017 study by da Costa and colleagues aimed to investigate the immediate effects of a single WBV session on the neuromuscular function of the quadriceps muscle and postural sway in individuals undergoing ACLR. Participants were randomly assigned to a control group (exercise therapy on a switched-off vibration platform) and an intervention group (exercise therapy on a switched-on vibration platform). Exercises on the switched-on vibration platform were performed at a frequency of 50 Hz and amplitude of 4 mm. Overall, WBV did not immediately alter quadriceps muscle function, electromyographic activity of the vastus muscles, or postural sway in ACLR patients. It appears that a single WBV session is insufficient to improve balance [19].

In the present study, dynamic postural control during a jump-landing task was examined, using a different balance test compared to the aforementioned studies. Additionally, participants in this study had to be at least 6 months post-surgery to enroll, with the intervention group averaging 13.47 ± 7.17 months and the control group averaging 10.00 ± 2.04 months post-surgery, indicating more chronic conditions compared to the mentioned studies.

A more critical comparison of prior findings indicates that, despite methodological variations, all previous studies employing multiple WBV sessions reported improvements in postural stability after ACLR. The only outlier was the study by da Costa et al., which implemented a single WBV session and observed no significant changes in neuromuscular or balance outcomes—suggesting that intervention dosage is a key factor in eliciting neuromuscular adaptations. Notably,

previous studies primarily assessed static or quasi-static balance using simple standing tasks on a force plate or Biodex system. In contrast, our study employed a dynamic jump-landing task, which more closely mirrors real-world athletic demands and situations where re-injury risk is elevated. Therefore, while our results align with the general consensus regarding WBV's efficacy, they may offer greater clinical and functional relevance by addressing dynamic postural control—a key determinant of safe return to sport following ACLR.

Improving TTS is clinically important. A recent video analysis of ACL injury mechanisms in basketball players estimated that injuries occur between 17 and 50 milliseconds after ground contact [52]. In contrast, a study comparing TTS in healthy individuals to those who have undergone ACLR found that the latter group required, on average, 110 milliseconds longer to stabilize. Prolonged TTS may be associated with impaired dynamic stability, which has been linked to higher injury risk in prior research. These findings underscore the importance of interventions aimed at improving jump-landing stability, both to prevent initial injuries and to reduce the risk of re-injury in individuals following ACLR [11].

Both groups showed significant improvements in YBT scores and hop test LSI, suggesting that adding WBV to routine exercises did not yield superior outcomes. These findings indicate that exercise therapy alone may be sufficient to enhance dynamic balance and physical performance in clinical assessments.Psychosocial factors, including kinesiophobia, may also influence neuromuscular performance and could help explain why improvements in YBT and LSI were similar across groups, highlighting the potential value of integrating psychological strategies alongside WBV to optimize functional recovery and return-to-sport outcomes.Kinesiophobia is an important factor in movement performance. Rostami et al. reported that individuals with ACL deficiency who exhibited higher levels of kinesiophobia showed lower peak vertical GRF during landing tasks, suggesting that these individuals may unload their injured limb due to fear of movement. However, the present study did not assess kinesiophobia, and therefore the potential influence of this factor on treatment outcomes and positive results remains unclear [53].

The exercise program in this study incorporated a wide range of sensorimotor elements, including bilateral and unilateral balance tasks, strength training, and exercises performed on unstable surfaces. These components are known to stimulate proprioceptors and enhance afferent sensory input to the central nervous system, thereby facilitating the development of more effective motor strategies [54]. Improved proprioceptive feedback is thought to contribute to enhanced postural control and neuromuscular coordination—key factors in dynamic performance. Recent reviews support the use of perturbation-based balance training as an effective approach for improving functional ability, joint stability, and inter-muscular coordination, particularly during high-demand tasks such as walking and jumping [55]. These neuromuscular adaptations may help explain, at least in part, the observed improvements in YBT and hop-test performance in the present study.

Additionally, the improvement observed in the physical performance and YBT might be attributed to the exercises' effects on reducing muscle inhibition. Although no direct study has specifically explored the impact of perturbation-based balance exercises on muscle inhibition, research on muscle strength and maximum torque suggests that combining these exercises with other forms of exercise, including strengthening, can be more effective in enhancing these measures [55]. After a ligament injury, the loss of sensory feedback from mechanical receptors disrupts the gamma loop, leading to quadriceps muscle weakness. As a result, individuals participating in these exercises focus on their weight distribution on the support surface, receiving tactile, verbal, and proprioceptive feedback, which may help restore gamma loop function. This could improve gamma loop feedback, reduce muscle inhibition, and enhance the quadriceps muscle's ability for dynamic knee stability [56].This improvement could be reflected in enhanced performance on the hop test and YBT.

Although the statistical analysis showed significant improvements in TTS, it is important to distinguish statistical significance from clinical relevance. Statistical significance indicates that an observed effect is unlikely to be due to chance, whereas clinical relevance relates to whether the magnitude of change is meaningful in a practical or therapeutic context. Because minimal clinically important difference (MCID) values have not yet been established for TTS, the minimal detectable change (MDC) was used as a reference to help determine whether the observed changes exceeded measurement

error and normal biological variability. MDC represents the smallest change that can be interpreted as a true improvement beyond random variation [57].

In this study, the intervention group demonstrated reductions in TTS of 262.2 ms in the mediolateral direction and 128.99 ms overall, both exceeding the MDC value of 123.8 ms previously reported in healthy individuals [58]. Although caution is warranted when extrapolating MDC thresholds derived from healthy populations to individuals following ACLR, this comparison suggests that the observed improvements are likely to represent changes beyond random measurement error and normal biological variability. Nevertheless, the clinical meaningfulness of these improvements should be interpreted cautiously, and future studies are required to establish ACLR-specific MDC or MCID values to more definitively determine the clinical relevance of TTS improvements following WBV-enhanced rehabilitation.

Although SI has higher reliability than TTS, the reductions in TTS exceeding the MDC suggest that these improvements likely reflect genuine enhancements in postural control rather than mere temporal fluctuations, even in the absence of significant changes in SI. This discrepancy can be explained by the fact that SI and TTS assess complementary but distinct aspects of dynamic stability: SI reflects inter-limb symmetry, whereas TTS captures the temporal component of achieving postural stabilization after a jump-landing task. Therefore, participants may achieve faster stabilization without substantial changes in limb symmetry, accounting for the unchanged SI values [20,32]. Together, these measures provide a more comprehensive understanding of postural control adaptations, and both should be considered when interpreting the results.

## 5. Limitations

This study has several methodological limitations that should be considered when interpreting the findings. One of the most important methodological limitations of this study is the absence of a WBV-only control group. Without such a group, it is not possible to determine whether the observed improvements in TTS and other outcomes are attributable solely to WBV, to the exercise program, or to a synergistic effect of the two interventions. Including a WBV-only group would have allowed for a clearer isolation of the independent effects of WBV and a more precise evaluation of its contribution to dynamic stability and postural control.Second, the absence of long-term follow-up assessments limits our understanding of the sustainability and clinical relevance of the observed treatment effects. This study evaluated outcomes only immediately post-intervention, leaving unanswered whether improvements in TTS and other functional measures persist over time, translate into successful return-to-sport, or reduce the risk of re-injury. Future studies should include 6–12 month follow-ups to assess re-injury rates, functional outcomes, and the durability of WBV effects, clarifying the long-term clinical impact of adding WBV to standard exercise protocols. Additionally, a longitudinal study with weekly evaluations could provide valuable insights into the recovery trajectory and clarify whether WBV accelerates progress toward desired outcomes, thereby enhancing our understanding of potential dose-response relationships. In the present study, jump-landing tasks were performed under non-fatigued conditions, limiting the applicability of findings to real-world sport scenarios where many ACL injuries occur under fatigue. Future studies should assess TTS and related outcomes during fatigued states to better evaluate the practical and clinical relevance of WBV for injury prevention in sport-specific contexts. A notable limitation of this study is the absence of assessments targeting the underlying mechanisms of the observed improvements. Specifically, EMG recordings during dynamic tasks such as jump-landing were not performed, and muscle strength or muscular coordination was not measured, limiting our understanding of the neuromuscular adaptations induced by WBV. Additionally, psychological factors such as kinesiophobia (fear of movement), which are well-established barriers to return-to-sport and full functional recovery after ACLR, were not evaluated. As such, improvements in outcomes like TTS may have been influenced by reductions in kinesiophobia or increases in self-confidence, which WBV could potentially modulate; however, these factors were not assessed, limiting interpretation of their contribution to the observed effects. Future studies should include EMG recordings, muscle strength and coordination assessments, and evaluations of psychological factors to better elucidate the mechanisms by which WBV—alone or combined with

other interventions—enhances neuromuscular performance and supports functional recovery.Moreover, WBV was initiated six months post-surgery. While this timing was reasonable for safety considerations, it falls just after the critical 3- to 6-month post-ACLR period, often regarded as the "golden window" for neuromuscular rehabilitation. Initiating WBV after this period may have missed the optimal opportunity to maximize neuromuscular adaptations, potentially attenuating the observed effects of the intervention on functional recovery.Additionally, blinding of participants was not feasible due to the perceptible nature of WBV, which may have introduced performance bias in effort-dependent outcomes such as the YBT. Treating physiotherapists were necessarily aware of group allocation because they administered the intervention, which may also contribute to performance bias. This limitation is acknowledged in the RoB 2 assessment. Furthermore, although the age range was restricted to control for confounding effects of aging, the sample consisted predominantly of young adults (mean age ≈ 32 years) who were recreationally active. Consequently, the findings may not be generalizable to older individuals following ACLR, to less physically active populations, or to elite athletes with substantially higher training loads and neuromuscular demands. In addition, the single-center design may limit external validity. Finally, sex-specific analyses were not performed despite well-documented sex differences in ACL injury risk and neuromuscular control. Future studies should adopt multicenter designs with broader age ranges and include participants across different activity levels, with sex-stratified analyses, to better elucidate how age-, sex-, and activity-related factors influence the effects of WBV following ACL reconstruction.

## 6. Conclusion

This study showed that adding WBV to routine exercise therapy produced selective effects, with additional improvements observed only in TTS during the jump-landing task in both the anterior-posterior and resultant directions. In contrast, WBV did not produce additional benefits for other outcomes. No between-group differences were found in YBT, or LSI, as both groups demonstrated similar improvements over time. Thus, the effects of WBV appear to be limited to TTS, with no added advantage over routine exercise for the remaining clinical or functional measures.

From a clinical perspective, these findings highlight the potential of WBV to enhance neuromuscular control in tasks that are highly dynamic and mechanically demanding, such as jump-landing. Such improvements are particularly relevant for athletes preparing to return to sport or other high-impact activities. Clinicians should consider the task-specific demands and neuromechanical sensitivity when integrating WBV into rehabilitation programs, recognizing that its advantages may be most pronounced in activities requiring rapid stabilization and fine-tuned postural control.

## Supporting information

**S1 Text. Study protocol (English version).**
(ZIP)

## Author contributions

**Conceptualization:** Roya Khanmohammadi.

**Data curation:** Zahra Rostampour.

**Formal analysis:** Zahra Rostampour.

**Methodology:** Roya Khanmohammadi.

**Resources:** Roya Khanmohammadi.

**Supervision:** Roya Khanmohammadi.

**Writing – original draft:** Zahra Rostampour.

**Writing – review & editing:** Roya Khanmohammadi.

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
