## [Decision Letter · Decision Letter 0]

18 Nov 2025

PLOS ONE

Dear Dr. Khanmohammadi,

Thank you for submitting your manuscript to PLOS ONE. After careful consideration, we feel that it has merit but does not fully meet PLOS ONE’s publication criteria as it currently stands. Therefore, we invite you to submit a revised version of the manuscript that addresses the points raised during the review process.

We look forward to receiving your revised manuscript.

Kind regards,

Esedullah Akaras

Academic Editor

PLOS ONE

**Journal Requirements:**

Reviewers' comments:

Reviewer's Responses to Questions

**Comments to the Author**

1. Is the manuscript technically sound, and do the data support the conclusions?

Reviewer #1: Yes

Reviewer #2: Yes

Reviewer #3: Yes

2. Has the statistical analysis been performed appropriately and rigorously?

Reviewer #1: Yes

Reviewer #2: Yes

Reviewer #3: I Don't Know

3. Have the authors made all data underlying the findings in their manuscript fully available?

Reviewer #1: Yes

Reviewer #2: Yes

Reviewer #3: Yes

4. Is the manuscript presented in an intelligible fashion and written in standard English?

Reviewer #1: Yes

Reviewer #2: Yes

Reviewer #3: Yes

Reviewer #1: The authors conducted a randomized trial to evaluate the effects of incorporating whole-body vibration into routine exercise on dynamic postural stability, dynamic balance and physical performance. The results showed that adding WBV to routine exercises led to greater improvements in jump-landing stability.

1. This study only includes participants aged 18 to 40 years, which is quite limited. Please justify this criterion. Also, this may also limit the generalizability.

2. With effect size of 0.066, the sample size of 15 per group seems quite low. Double check the calculation is recommended. Also, please justify the selectin of 0.066 and 0.081.

3. The statistical significance for TTS improvement was found. Please discuss its clinical significance and relevance.

Reviewer #2: Dear editors,

This randomized controlled trial (RCT) investigates a clinically relevant question using a task-specific outcome (jump-landing stability) in a post-ACL reconstruction population. The study is well-structured, registered, and assessor-blinded. However, critical methodological flaws, statistical inconsistencies, and overinterpretations compromise its validity and clinical interpretability. Below is a line-by-line, rigorous scientific critique with specific comments, contradictions, and mandatory corrections.

Title

Comment:

• The title is somewhat long and could be more concise. The phrase “for improving jump-landing stability post-ACL reconstruction” is grammatically correct but stylistically heavy.

Abstract

• The effect size (η²) is not mentioned; this is essential in RCTs.

• The phrase “clinical tests” is vague; specify the tools (e.g., Y-Balance Test and Limb Symmetry Index).

• The final sentence “suggesting enhanced neuromuscular control…” overstates the findings; it should read:

“…suggesting potential enhancement in neuromuscular control…”

• The phrase “TTS significantly improved” is misleading, as TTS is a time-based measure where lower values represent better performance. Therefore, the correct expression should be “TTS significantly decreased.” Use this phrasing consistently throughout the manuscript to ensure conceptual accuracy.

• The text reports “P < 0.017” for YBT/LSI, implying a Bonferroni correction for three YBT directions. However, the Results section (Line 359) reports “P ≤ 0.016.” This inconsistency must be addressed by standardizing the threshold to P < 0.016 or by providing a clear justification for using 0.017.

• The statement “routine exercise alone may be sufficient” overgeneralizes the findings. Although both groups improved significantly (P < 0.001), this reflects descriptive equivalence rather than evidence of sufficiency. Rephrase the sentence as: “No additional benefit was observed from WBV compared to routine exercise.”

Introduction

• The opening section (lines 58–70) is overly descriptive with many statistics that do not directly support the research gap. Consider condensing it.

• In line 93, “An important methodological limitation” has a capitalization error (“An” should start a new sentence or be connected grammatically).

• The research gap should be explicitly stated at the end, for example:

“However, the effect of WBV on dynamic postural stability during sport-specific tasks such as jump-landing remains unclear.”

• Reviews fatigue-induced changes in neuromuscular/biomechanical variables post-ACLR, showing similar responses to controls but persistent deficits—aligns with Lines 71–78 (re-injury risk) and 98–100 (dynamic tasks underestimate deficits). The authors may cite the following reference in lines 71–78 or 98-100: Effect of Fatigue on Neuromuscular and Biomechanical Variables After Anterior Cruciate Ligament Reconstruction: A Systematic Review. For example, Persistent neuromuscular deficits, exacerbated by fatigue [cite],

• “Effect of Fatigue on Neuromuscular and Biomechanical Variables After Anterior Cruciate Ligament Reconstruction: A Systematic Review”

• Discussion (Lines 374–559) : This reference supports the discussion on quadriceps inhibition and coordination (lines 388–396). Fatigue often mirrors patterns in healthy individuals but reveals ACL-specific vulnerabilities in landing kinetics. Suggested integration: “Our TTS improvements may mitigate fatigue-related biomechanical alterations, as fatigue does not differentially impair ACLR patients vs. controls [cite], suggesting WBV enhances resilience under load.”

• Limitations (Lines 527–559): Since no fatigue protocol was tested in the current study, this reference can justify future research incorporating fatigue during jump-landing tasks.

Suggested integration: “Future work should incorporate fatigue protocols [cite] to assess WBV’s durability in real-world athletic demands.”

The authors can used the following reference in different part of study; Psychosocial Interventions Seem to Reduce Kinesiophobia After Anterior Cruciate Ligament Reconstruction But Higher Level of Evidence Is Needed: A Systematic Review and Meta-Analysis

Introduction (Lines 58–132 ): Meta-analyses indicate that psychosocial interventions can reduce kinesiophobia, a major barrier to return-to-sport (Lines 66–71, 105–107). This aligns with the equivocal findings for WBV interventions (Lines 89–92). Kinesiophobia, as a key predictor of delayed or incomplete return-to-sport, may persist despite ACL reconstruction, highlighting the potential value of adjunctive therapies such as WBV for a more holistic recovery.

Discussion (Lines 374–559): Psychosocial factors influence neuromuscular control, which may explain why YBT and LSI improvements were similar across groups. While WBV effectively targets biomechanical stability, integrating psychosocial strategies could better address kinesiophobia, potentially enhancing return-to-sport rates.

Limitations (Lines 527–565): Evidence for long-term benefits of psychosocial interventions remains limited, and no long-term follow-up was conducted (Lines 538–549). Future research should combine WBV with psychological support in robust longitudinal trials to evaluate the sustained impact on both neuromuscular performance and psychological readiness.

• Line 89–92: The manuscript states that da Costa et al. found “no significant effects,” but this is misleading. That study examined the acute, single-session effects of WBV, whereas the present study investigates a multi-session intervention. The distinction between acute and long-term designs must be made explicit. Revise to: “Unlike the acute, single-session WBV protocol used by da Costa et al., the current study implemented a multi-session intervention.”

• Line 97–98 :The phrase “static tasks impose relatively low neuromuscular demands” is inaccurate. Tasks performed on the Biodex platform at Levels 1–2 are inherently unstable and challenge balance substantially. This is a false generalization. Revise to: “Static balance tasks on low Biodex stability levels may not fully replicate the multi-planar and high-velocity neuromuscular demands of sport.”

• Line 121–122: The text cites studies reporting reduced GRF [28] and shortened TTS [29] in healthy participants as if these findings apply to post-ACL populations. This is an overextension. These results highlight a research gap, not supporting evidence. Rephrase to: “Preliminary evidence in healthy populations suggests that WBV may reduce GRF and TTS; however, its effects in post-ACL individuals remain unclear.”

Other studies that the authors could reference to strengthen the Introduction and Discussion include:

• Evaluation of Potential Factors Affecting Knee Function in Athletes After Anterior Cruciate Ligament Reconstruction

• Relationship Between Kinesiophobia and Vertical Ground Reaction Force in Anterior Cruciate Ligament Reconstructed and Deficient Patients During Landing Task

• Hip Abductor and Adductor Muscles Activity Patterns During Landing After Anterior Cruciate Ligament Injury

Methods

• Explain the allocation concealment and blinding procedures. Who administered WBV? Was the therapist blinded to group assignment?

• The reported effect size = 0.066 is quite small; justification is needed (details from the pilot study should be included).

• It is unclear why SI is treated as the primary outcome when the main results focus on TTS. Clarify this inconsistency.

• Repeated-measures ANOVA is appropriate, but assumptions (sphericity, homogeneity) should be reported.

• Technical details are accurate but require methodological references (e.g., Wikstrom et al., 2005).

• Frequency (Hz), amplitude (mm), and duration per session must be explicitly stated for reproducibility.

• Control group equivalence: Both groups performed “routine exercises,” but intensity and duration may differ; a comparative table should be added.

• Line 530–536 : A major methodological limitation is the absence of a WBV-only group. Without this comparison, it is impossible to determine whether the observed effects result from WBV itself or from its interaction with routine exercises. This design prevents answering a key question: Is WBV alone sufficient to produce similar improvements?

• Line 175–178 : Although initiating WBV six months post-operation is justified for safety reasons, this timing misses the critical neuromuscular recovery window typically observed between 3 and 6 months after ACL reconstruction. This delay may reduce the intervention’s potential impact on neuromuscular adaptation and functional restoration.

Line 231–232: The use of a 3-second window for TTS analysis is questionable. Although references [20,31] are cited, ACL injuries typically occur within less than 50 milliseconds [50]. A 3-second post-landing analysis may not accurately reflect the critical phase of dynamic stabilization relevant to ACL injury mechanisms. The authors should justify the ecological validity of this time frame or consider restricting the analysis to the initial 0–1 second after landing.

Line 252 : The equation RVTTS = √(MLTTS² + APTTS²) assumes that both TTS components share the same units and scale, which may not be valid. TTS values should be normalized—such as by jump height, body weight, or another standard measure—before vector summation. Without normalization, this calculation is mathematically and conceptually invalid.

Line 208–210 : The exclusive use of a diagonal jump is justified with reference to Patterson [9]; however, this choice limits generalizability. Forward jump-landing tasks are more commonly associated with ACL injury mechanisms. The authors should explain why forward or vertical jumps were excluded or provide a rationale for choosing the diagonal direction as representative of sport-specific demands.

Line 311: The protocol applies WBV before the main exercise session, but no rationale is provided for this sequencing. Delivering WBV first may induce fatigue or neuromuscular pre-activation that could confound subsequent performance in strength and balance tasks. The authors should justify this order or discuss its potential impact on training outcomes.

Table 3:The Week 4 protocol (50 Hz with nine exercises) represents a relatively high cumulative WBV exposure. Such intensity may increase the risk of neuromuscular fatigue, overuse, or diminishing adaptive returns. The authors should provide justification for this progression, supported by safety guidelines or prior studies demonstrating the tolerability of similar dosing parameters.

Results

• Some results are redundantly presented in both text and tables — shorten the text to avoid repetition.

• p-values should be reported consistently (e.g., use p < 0.001 instead of p = 0.001).

• Add 95% confidence intervals for between-group differences.

• Interpretation should acknowledge that significant effects were found only for AP and resultant directions in TTS, not all directions. This nuance should appear in the

• Line 365–367: The Limb Symmetry Index (LSI) formula appears reversed. The manuscript uses (Healthy limb time / Injured limb time) × 100, which incorrectly yields values greater than 100% when the injured limb performs better—a conceptual error. The standard and widely accepted formula is: LSI = (Injured limb performance / Healthy limb performance) × 100. This formulation ensures that values below 100% correctly indicate performance deficits in the injured limb. All LSI values must be recalculated.

• 351–352 : ML and V TTS data omitted | Only "no significant effects" → raw means ± SD must be in table. Transparency violation.

• 339–340 : Effect size underreported | AP TTS: η² = 0.568 (WBV group) vs. 0.150 (control) → large difference, but not in abstract. |

Discussion.

Line 387–388: The statement suggesting that the findings “may reflect quadriceps inhibition or coordination deficits” is speculative and unsupported by the data. Since no EMG, strength, or neural inhibition measures were collected, such interpretations exceed the study’s evidence base. These claims should be softened or removed to avoid overinterpretation.

Line 396–403: The citation of Maghbouli et al. as supportive evidence is inappropriate, as that meta-analysis examined torque outcomes rather than TTS. The connection between muscle torque improvements and time-to-stabilization is indirect at best. Clarify this distinction and avoid implying direct comparability between unrelated outcome measures.

Line 492–498: The statement “extended TTS may compromise injury prevention” suggests a causal relationship not established in this study. TTS is a correlational performance indicator, not a validated causal predictor of re-injury risk. Rephrase to emphasize association rather than causation, e.g., “Prolonged TTS may be associated with impaired dynamic stability, which has been linked to higher injury risk in prior research.”

Line 560–565: The claim that “WBV is more effective than exercise alone” is overstated. The observed benefit was limited to TTS outcomes, with no significant differences in SI, YBT, or LSI measures. The conclusion should be revised to accurately reflect the selective nature of the effect, for example: “WBV yielded additional improvements in TTS compared with exercise alone, but no differences were observed in SI, YBT, or LSI.”

Reviewer #3: Although the design of the article is good, there are important issues that are overlooked in the protocol that need to be clarified.

1- Provide a study timeline.

2- In the control group, how was the initial intensity of the exercises (10RM) measured? You got 10RM with a specific method.

3- What was the basis for increasing the intensity of the exercises in the intervention group? What was the basis for changing the frequency and duration of vibration administration?

4- What has been the effectiveness of your protocol in improving proprioception? Has proprioception been assessed at all? With what method and with what test?

**Do you want your identity to be public for this peer review?** For information about this choice, including consent withdrawal, please see our Privacy Policy

Reviewer #1: No

Reviewer #2: No

Reviewer #3: No

---

## [Author Response · Author response to Decision Letter 1]

29 Nov 2025

Dear Reviewer 1

We sincerely appreciate your thorough review of the manuscript and the valuable comments and suggestions provided. We believe that the manuscript has significantly improved as a result of incorporating your feedback, which has been highlighted in red throughout the text.

Question 1: This study only includes participants aged 18 to 40 years, which is quite limited. Please justify this criterion. Also, this may also limit the generalizability.

Answer 1: Thank you for raising this point. We specifically restricted our sample to 18–40 years in order to control a critical confounding variable: age, which is strongly associated with declines in balance, postural stability, and neuromuscular control. Evidence shows that as people age, there is an increase in postural sway, slower sensory integration, and reduced reflexive control of balance [1]. By limiting the age range, we reduced the risk that age-related deficits in balance would mask or distort the effects of our intervention on functional and biomechanical outcomes, such as jump biomechanics and postural control. Additionally, keeping the age range narrower helped us obtain a more homogeneous sample in terms of healing potential, physical activity level, and neuromuscular adaptation, improving internal validity. We acknowledge, however, that this choice limits generalizability.

1. Slobounov SM, Haibach PS, Newell KMJERoA, Activity P. Aging-related temporal constraints to stability and instability in postural control. 2006;3(2):55-62.

Question 2: With effect size of 0.066, the sample size of 15 per group seems quite low. Double check the calculation is recommended. Also, please justify the selecting of 0.066 and 0.081.

Answer 2: Please note that the reported effect size (0.066) is based on η², not Cohen’s d. According to conventional benchmarks, η² of ~0.01 is considered small, ~0.06 medium, and ~0.14 large. Thus, our estimated η² of 0.066 falls in the medium range. Moreover, these values come from our pilot study of 10 participants (5 per group), and we have now added the full pilot data details into the Methods section for transparency. Below is a screenshot of G*Power 3.1.3 showing the input parameters (η² = 0.066, power = 0.80, α = 0.05, ε = 1, correlation = 0.50).

“The sample size was determined using G*Power 3.1.3 based on a pilot study of 10 participants (5 per group), focusing on the SI and TTS in the AP direction. In that pilot, the AP SI changed from 0.035 to 0.030 (–0.005) in the control group and from 0.042 to 0.036 (–0.006) in the intervention group; the AP TTS changed from 950 ms to 900 ms (–50 ms) in control and from 960 ms to 854 ms (–106 ms) in the intervention group. The calculation used a within between interaction effect size (η²) of 0.066 and 0.081, a power of 0.80, α = 0.05, a non sphericity correction (ε) = 1, and a correlation among repeated measures of 0.50. The analysis indicated that at least 30 participants were needed to detect a within between interaction in a 2 group, 2 measurement design with those parameters. Consequently, 34 participants were recruited to account for a potential 10% dropout rate.”

Question 3: The statistical significance for TTS improvement was found. Please discuss its clinical significance and relevance.

Answer 3: We added the following section to address your comment:

“Although the statistical analysis revealed significant improvements in TTS, it is essential to distinguish between statistical significance and clinical relevance. Statistical significance reflects the probability that an observed effect is not due to chance, whereas clinical relevance pertains to whether the magnitude of change is meaningful in a practical or therapeutic context. Although minimal clinically important difference (MCID) thresholds have not yet been established for TTS, the minimal detectable change (MDC) provides a useful benchmark for determining whether observed changes exceed measurement error and normal biological variability. MDC represents the smallest change that can be interpreted as a real improvement beyond random variation [2].

In this study, the intervention group demonstrated reductions in TTS of 262.2 ms in the mediolateral direction and 128.99 ms overall, both exceeding the MDC value of 123.8 ms previously reported in healthy individuals [3]. Although caution is warranted when extrapolating MDC values from healthy populations to post-ACLR patients, these findings strongly suggest that the observed improvements reflect true physiological and neuromuscular adaptations rather than random measurement error, thereby supporting the effectiveness of the WBV-enhanced rehabilitation protocol.”

Dear Reviewer 2

We sincerely appreciate your thorough review of the manuscript and the valuable comments and suggestions provided. We believe that the manuscript has significantly improved as a result of incorporating your feedback, which has been highlighted in blue throughout the text.

Title

Question 1: The title is somewhat long and could be more concise. The phrase “for improving jump- landing stability post-ACL reconstruction” is grammatically correct but stylistically heavy.

Answer 1: We modified the title to “Effect of whole-body vibration combined with exercise therapy on jump-landing stability after ACL reconstruction: A randomized controlled trial”

Abstract

Question 2: The effect size (η2) is not mentioned; this is essential in RCTs.

Answer 2: The effect size (η2) was added to the abstract.

Question 3: The phrase “clinical tests” is vague; specify the tools (e.g., Y-Balance Test and Limb Symmetry Index).

Answer 3: We have replaced the vague term “clinical tests” with the specific outcome measures used in the study—namely, the Y-Balance Test (YBT) and the Limb Symmetry Index (LSI)—to improve clarity.

Question 4: The final sentence “suggesting enhanced neuromuscular control...” overstates the findings; it should read: “...suggesting potential enhancement in neuromuscular control...”

Answer 4: Based on your subsequent feedbacks, the sentence “suggesting enhanced neuromuscular control…” has been entirely removed from the manuscript.

Question 5: The phrase “TTS significantly improved” is misleading, as TTS is a time-based measure where lower values represent better performance. Therefore, the correct expression should be “TTS significantly decreased.” Use this phrasing consistently throughout the manuscript to ensure conceptual accuracy.

Answer 5: We have revised the manuscript to replace all instances of “TTS significantly improved” with “TTS significantly decreased”.

Question 6: The text reports “P < 0.017” for YBT/LSI, implying a Bonferroni correction for three YBT directions. However, the Results section (Line 359) reports “P ≤ 0.016.” This inconsistency must be addressed by standardizing the threshold to P < 0.016 or by

providing a clear justification for using 0.017.

Answer 6: We respectfully note a misunderstanding regarding the reported P-values. In our study, for the YBT, the ANOVA results for each direction were as follows: A YBT: P = 0.001, PM YBT: P = 0.016, and PL YBT: P < 0.001. Based on these values, we reported P ≤ 0.016 in the main text as a summary of the ANOVA results across directions. No Bonferroni correction was applied, because these P-values are from the main effect of time in ANOVA, not from post hoc pairwise comparisons, and for a two-level factor (pre vs. post) Bonferroni correction is not required. For the LSI at 6 months, the ANOVA result was P < 0.001, reflecting a significant main effect of time. In summary, the reported P-values in the main text and abstract accurately reflect the ANOVA results, and P ≤ 0.016 is equivalent to P < 0.017, so both represent the same statistical significance. We have now standardized the presentation to P ≤ 0.016 throughout the manuscript for consistency.

Question 7: The statement “routine exercise alone may be sufficient” overgeneralizes the findings. Although both groups improved significantly (P < 0.001), this reflects descriptive equivalence rather than evidence of sufficiency. Rephrase the sentence as: “No additional benefit was observed from WBV compared to routine exercise.”

Answer 7: We would like to clarify that our study included two groups: one group received routine exercise combined with WBV, and the other group performed routine exercise alone. Therefore, our comparison was between “routine exercise + WBV” and “routine exercise only,” not between WBV and routine exercise.

In line with the reviewer’s comment, we have revised the sentence to avoid any overgeneralization. The revised text now reads:

“Overall, the study showed that adding WBV to routine exercises produced selective benefits, with additional improvements observed only in TTS during the jump-landing task. In contrast, both groups demonstrated comparable improvements in YBT and LSI, with no between-group differences. These findings indicate that WBV did not provide added benefit beyond exercise alone for these latter outcomes.”

Introduction

Question 8: The opening section (lines 58–70) is overly descriptive with many statistics that do not directly support the research gap. Consider condensing it.

Answer 8: We have condensed the opening section, reducing the number of lines and statistics by approximately half to focus more directly on the research gap.

Question 9: In line 93, “An important methodological limitation” has a capitalization error (“An” should start a new sentence or be connected grammatically).

Answer 9: We revised it.

Question 10: The research gap should be explicitly stated at the end, for example:

“However, the effect of WBV on dynamic postural stability during sport-specific tasks

such as jump-landing remains unclear.”

Answer 10: This sentence has been added at line 120, immediately before stating the study objectives and hypotheses, to clearly articulate the research gap.

Question 11: Reviews fatigue-induced changes in neuromuscular/biomechanical variables post-ACLR, showing similar responses to controls but persistent deficits—aligns with Lines 71–78 (re-injury risk) and 98–100 (dynamic tasks underestimate deficits). The authors may cite the following reference in lines 71–78 or 98-100: Effect of Fatigue on Neuromuscular and Biomechanical Variables After Anterior Cruciate Ligament Reconstruction: A Systematic Review. For example, Persistent neuromuscular deficits, exacerbated by fatigue [cite],

Answer 11: We incorporated this point into the manuscript by adding a sentence highlighting the role of fatigue. Specifically, we noted that “Moreover, evidence indicates that these neuromuscular deficits may become even more pronounced under fatigue, further compromising postural control and increasing the likelihood of re-injury in individuals after ACL reconstruction [13, cite added].”

Question 12: A) Discussion (Lines 374–559) : This reference supports the discussion on quadriceps inhibition and coordination (lines 388–396). Fatigue often mirrors patterns in healthy individuals but reveals ACL-specific vulnerabilities in landing kinetics. Suggested

integration: “Our TTS improvements may mitigate fatigue-related biomechanical

alterations, as fatigue does not differentially impair ACLR patients vs. controls [cite],

suggesting WBV enhances resilience under load.”

B) Limitations (Lines 527–559): Since no fatigue protocol was tested in the current study, this reference can justify future research incorporating fatigue during jump-landing tasks.

Suggested integration: “Future work should incorporate fatigue protocols [cite] to assess WBV’s durability in real-world athletic demands.”

Answer 12: A) Thank you for your comment. In response, we have added the following sentences to the “Discussion”: “Furthermore, emerging evidence indicates that fatigue often induces similar neuromuscular and biomechanical responses in ACLR patients and healthy individuals; however, it can also reveal ACL-specific vulnerabilities during landing tasks [13, cite added]. In this context, the observed improvements in TTS may suggest enhanced resilience against fatigue-related biomechanical alterations, potentially supporting more stable performance under load.”

B) We added this section to the “Limitations”: “In the present study, jump-landing tasks were not performed under fatigued conditions which limit our insight into WBV’s impact during realistic athletic demands. Future research should incorporate fatigue protocols in dynamic tasks to evaluate the durability and practical benefits of WBV in sport-specific scenarios.”

Question 13: A) The authors can used the following reference in different part of study; Psychosocial Interventions Seem to Reduce Kinesiophobia After Anterior Cruciate Ligament Reconstruction But Higher Level of Evidence Is Needed: A Systematic Review and Meta-Analysis

Introduction (Lines 58–132 ): Meta-analyses indicate that psychosocial interventions can reduce kinesiophobia, a major barrier to return-to-sport (Lines 66–71, 105–107). This aligns with the equivocal findings for WBV interventions (Lines 89–92). Kinesiophobia, as a key predictor of delayed or incomplete return-to-sport, may persist despite ACL reconstruction, highlighting the potential value of adjunctive therapies such as WBV for a more holistic recovery.

B) Discussion (Lines 374–559): Psychosocial factors influence neuromuscular control, which may explain why YBT and LSI improvements were similar across groups. While WBV effectively targets biomechanical stability, integrating psychosocial strategies could better address kinesiophobia, potentially enhancing return-to-sport rates.

C) Limitations (Lines 527–565): Evidence for long-term benefits of psychosocial

interventions remains limited, and no long-term follow-up was conducted (Lines 538–

549). Future research should combine WBV with psychological support in robust

longitudinal trials to evaluate the sustained impact on both neuromuscular performance

and psychological readiness.

Answer 13: A) We added this section to the “Introduction”: “Persistent fear of movement (kinesiophobia) is another factor that may hinder return-to-sport and limit functional recovery after ACL reconstruction. Meta-analyses indicate that psychosocial interventions can reduce kinesiophobia, which remains a major barrier even after surgery [14].

Together, these findings suggest that persistent deficits in postural control and unresolved kinesiophobia are key contributors to limited functional recovery and increased vulnerability to re-injury. This highlights the importance of incorporating targeted interventions into rehabilitation programs to enhance both postural stability and patient confidence, thereby reducing the risk of subsequent injury.”

B) We added this section to the “Discussion”: “Psychosocial factors, including kinesiophobia, may also influence neuromuscular performance and could help explain why improvements in YBT and LSI were similar across groups, highlighting the potential value of integrating psychological strategies alongside WBV to optimize functional recovery and return-to-sport outcomes.”

C) We added this section to the “Limitations”: “Moreover, this study did not evaluate psychological factors, despite the well-established role of kinesiophobia (fear of movement) as a major barrier to return-to-sport and full functional recovery after ACL reconstruction. Future research should examine both the potential effects of WBV on kinesiophobia and the combined effects of WBV with interventions targeting psychological factors in robust longitudinal trials to assess their sustained impact on neuromuscular performance and psychological readiness.”

Question 14: Line 89–92: The manuscript states that da Costa et al. found “no significant effects,” but this is misleading. That study examined the acute, single-session effects of WBV, whereas the present study investigates a multi-session intervention. The distinction between acute and long-term designs must be made explicit. Revise to: “Unlike the acute, single-session WBV protocol used by da Costa et al., the current study

implemented a multi-session intervention.”

Answer 14: In the Introduction, we clarified that da Costa et al. examined an acute, single-session WBV protocol, whereas othe

---

## [Decision Letter · Decision Letter 1]

23 Dec 2025

Dear Dr. Khanmohammadi,

Thank you for submitting your manuscript to PLOS ONE. After careful consideration, we feel that it has merit but does not fully meet PLOS ONE’s publication criteria as it currently stands. Therefore, we invite you to submit a revised version of the manuscript that addresses the points raised during the review process.

We look forward to receiving your revised manuscript.

Kind regards,

Esedullah Akaras

Academic Editor

PLOS One

**Journal Requirements:**

Reviewers' comments:

Reviewer's Responses to Questions

**Comments to the Author**

Reviewer #1: (No Response)

Reviewer #2: All comments have been addressed

Reviewer #3: (No Response)

2. Is the manuscript technically sound, and do the data support the conclusions?

Reviewer #1: (No Response)

Reviewer #2: Yes

Reviewer #3: (No Response)

3. Has the statistical analysis been performed appropriately and rigorously?

Reviewer #1: (No Response)

Reviewer #2: Yes

Reviewer #3: (No Response)

4. Have the authors made all data underlying the findings in their manuscript fully available?

Reviewer #1: (No Response)

Reviewer #2: Yes

Reviewer #3: (No Response)

5. Is the manuscript presented in an intelligible fashion and written in standard English?

Reviewer #1: (No Response)

Reviewer #2: Yes

Reviewer #3: (No Response)

Reviewer #1: (No Response)

Reviewer #2: Overall Evaluation:

This revised manuscript (R1) reports an RCT examining the effects of whole-body vibration combined with routine exercise on postural stability, balance, and physical performance in athletes after ACL reconstruction. The study is relevant, uses objective biomechanical outcomes, and includes pilot-based power calculations. Revisions addressed prior reviewer comments (sample size, TTS reporting, introduction). Some methodological and reporting issues remain, such as incomplete blinding details, unstandardized training load, and limited statistical reporting, but overall, the manuscript is well-improved and requires only minor revision before being suitable for publication.

Reviewer #3: (No Response)

**Do you want your identity to be public for this peer review?** For information about this choice, including consent withdrawal, please see our Privacy Policy

Reviewer #1: No

Reviewer #2: No

Reviewer #3: No

---

## [Author Response · Author response to Decision Letter 2]

25 Dec 2025

Dear Reviewer 2

We sincerely appreciate your thorough review of the manuscript and the valuable comments and suggestions provided. We believe that the manuscript has significantly improved as a result of incorporating your feedback, which has been highlighted in red throughout the text. In addition, some comments that addressed overlapping or closely related issues were integrated and addressed together to avoid redundancy and improve clarity.

Question 1: lines ~180–200: The description of blinding remains insufficient. While the authors indicate a single-blind design (assessor-blinded), it is unclear whether participants or WBV therapists were blinded and how this was achieved (e.g., sham or placebo WBV). In physical intervention RCTs, participant blinding is challenging but essential to reduce performance bias, especially for subjective outcomes such as Y-Balance Test scores. Allocation concealment is not explicitly described—who generated the random sequence and how was it concealed from recruiters (e.g., sealed envelopes or third-party randomization)? Please provide the exact protocol and discuss the risk of bias using RoB 2 in the limitations. Without this, internal validity is compromised.

Answer 1: We revised this section to: “Participants were randomly assigned to either the intervention group (WBV plus routine exercise) or the control group (routine exercise alone) with a 1:1 allocation ratio. Random sequence generation was performed by an independent researcher using a web-based randomization service (www.randomization.com). Group allocation was concealed using sequentially numbered, opaque, sealed envelopes prepared by a third party not involved in participant recruitment, intervention delivery, or outcome assessment.

The study employed a single-blind (assessor-blinded) design. Interventions were delivered by a trained physiotherapist who was not involved in outcome assessment. Outcome measurements were performed by an independent assessor who remained blinded to group allocation to minimize detection bias. Blinding of participants was not feasible due to the perceptible mechanical stimuli produced by WBV, which cannot be convincingly replicated without an active platform, and no validated sham WBV protocol was available. Treating physiotherapists were necessarily aware of group allocation because they administered the intervention.”

Additionally, the following was added to the Limitations section: “Additionally, blinding of participants was not feasible due to the perceptible nature of WBV, which may have introduced performance bias in effort-dependent outcomes such as the YBT. Treating physiotherapists were necessarily aware of group allocation because they administered the intervention, which may also contribute to performance bias. This limitation is acknowledged in the RoB 2 assessment.”

Question 2: lines ~450–500: The age range (18–40) is justified for homogeneity, but the sample is mostly young (~25 years) elite athletes, limiting applicability to older ACLR patients or recreational athletes. Discuss how age-related neuromuscular declines (e.g., proprioception) might interact with WBV. Single-center design and lack of sex stratification (despite sex differences in ACL injury) are weaknesses—add in limitations and suggest multicenter studies.

Answer 2: We added the following to the Limitation section: “Furthermore, although the age range was restricted to control for confounding effects of aging, the sample consisted predominantly of young adults (mean age ≈32 years) who were recreationally active. Consequently, the findings may not be generalizable to older individuals following ACLR, to less physically active populations, or to elite athletes with substantially higher training loads and neuromuscular demands. In addition, the single-center design may limit external validity. Finally, sex-specific analyses were not performed despite well-documented sex differences in ACL injury risk and neuromuscular control. Future studies should adopt multicenter designs with broader age ranges and include participants across different activity levels, with sex-stratified analyses, to better elucidate how age-, sex-, and activity-related factors influence the effects of WBV following ACL reconstruction.”

Additionally, we revised the Participant section to: “The sample was restricted to individuals aged 18–40 years to control for age as a key confounding factor associated with declines in balance, postural stability, and neuromuscular control. With advancing age, postural sway increases, sensory integration slows, and proprioceptive as well as reflexive balance control deteriorate [31]. Such age-related neuromuscular changes may alter responsiveness to WBV, potentially affecting the magnitude and nature of postural and functional adaptations in older compared with younger individuals. By limiting the age range, we reduced the likelihood that age-related deficits would obscure or modify the observed effects of WBV on postural control and jump biomechanics.”

Question 3: lines ~400–450: Statistically significant TTS reductions are reported, but clinical relevance without minimal detectable change (MDC) thresholds for ACLR patients may be overstated. MDC = 123.8 ms comes from healthy individuals—how is this applied here? Provide context: e.g., “TTS reduction of 128.99 ms exceeds MDC, indicating meaningful improvement, though confirmation in ACLR populations is needed.”

Answer 3: These paragraphs are intended to convey your comment: “Although the statistical analysis showed significant improvements in TTS, it is important to distinguish statistical significance from clinical relevance. Statistical significance indicates that an observed effect is unlikely to be due to chance, whereas clinical relevance relates to whether the magnitude of change is meaningful in a practical or therapeutic context. Because minimal clinically important difference (MCID) values have not yet been established for TTS, the minimal detectable change (MDC) was used as a reference to help determine whether the observed changes exceeded measurement error and normal biological variability. MDC represents the smallest change that can be interpreted as a true improvement beyond random variation [57].

In this study, the intervention group demonstrated reductions in TTS of 262.2 ms in the mediolateral direction and 128.99 ms overall, both exceeding the MDC value of 123.8 ms previously reported in healthy individuals [58]. Although caution is warranted when extrapolating MDC thresholds derived from healthy populations to individuals following ACLR, this comparison suggests that the observed improvements are likely to represent changes beyond random measurement error and normal biological variability. Nevertheless, the clinical meaningfulness of these improvements should be interpreted cautiously, and future studies are required to establish ACLR-specific MDC or MCID values to more definitively determine the clinical relevance of TTS improvements following WBV-enhanced rehabilitation.”

Question 4: One of the most important methodological limitations is the absence of a control group receiving only WBV; without this group, it is impossible to determine whether the observed improvement in TTS is due to WBV itself or a synergistic effect of its combination with exercise.

Answer 4: This limitation has been added to the Limitations section: “One of the most important methodological limitations of this study is the absence of a WBV-only control group. Without such a group, it is not possible to determine whether the observed improvements in TTS and other outcomes are attributable solely to WBV, to the exercise program, or to a synergistic effect of the two interventions. Including a WBV-only group would have allowed for a clearer isolation of the independent effects of WBV and a more precise evaluation of its contribution to dynamic stability and postural control.”

Question 5: Although this period is reasonable for safety, the 3- to 6-month post-surgery phase is recognized as the "golden window" for neuromuscular rehabilitation.

Starting WBV after 6 months may have missed the optimal opportunity to induce neuromuscular changes, and therefore the effects of WBV might appear less than potentially possible.

Answer 5: This limitation has been added to the Limitations section: “Moreover, WBV was initiated six months post-surgery. While this timing was reasonable for safety considerations, it falls just after the critical 3- to 6-month post-ACLR period, often regarded as the “golden window” for neuromuscular rehabilitation. Initiating WBV after this period may have missed the optimal opportunity to maximize neuromuscular adaptations, potentially attenuating the observed effects of the intervention on functional recovery.”

Question 6: The lack of EMG use during landing, failure to measure muscle strength or muscular coordination, and the absence of an assessment of kinesiophobia or other psychological factors represent a significant gap in interpreting the underlying mechanisms. An improvement in TTS might be due to a reduction in fear of movement or increased self-confidence, which WBV could indirectly influence, but this aspect was not investigated. This may limit statistical power to detect smaller differences, especially in outcomes like SI, which did not show a significant change.

Answer 6: This limitation has been added to the Limitations section: “A notable limitation of this study is the absence of assessments targeting the underlying mechanisms of the observed improvements. Specifically, EMG recordings during dynamic tasks such as jump-landing were not performed, and muscle strength or muscular coordination was not measured, limiting our understanding of the neuromuscular adaptations induced by WBV. Additionally, psychological factors such as kinesiophobia (fear of movement), which are well-established barriers to return-to-sport and full functional recovery after ACLR, were not evaluated. As such, improvements in outcomes like TTS may have been influenced by reductions in kinesiophobia or increases in self-confidence, which WBV could potentially modulate; however, these factors were not assessed, limiting interpretation of their contribution to the observed effects. Future studies should include EMG recordings, muscle strength and coordination assessments, and evaluations of psychological factors to better elucidate the mechanisms by which WBV—alone or combined with other interventions—enhances neuromuscular performance and supports functional recovery.”

Question 7: This study only examined short-term effects (immediately after the intervention); the lack of a 6- or 12-month follow-up to assess the sustainability of improvements, return-to-sport rate, or risk of re-injury is a major clinical weakness. Does the improvement in TTS translate into a reduced risk of re-injury? Without follow-up, there is no answer to this question.

Answer 7: This limitation has been added to the Limitations section: “Second, the absence of long-term follow-up assessments limits our understanding of the sustainability and clinical relevance of the observed treatment effects. This study evaluated outcomes only immediately post-intervention, leaving unanswered whether improvements in TTS and other functional measures persist over time, translate into successful return-to-sport, or reduce the risk of re-injury. Future studies should include 6–12 month follow-ups to assess re-injury rates, functional outcomes, and the durability of WBV effects, clarifying the long-term clinical impact of adding WBV to standard exercise protocols.”

Question 8: The landing task was performed under non-fatigued conditions, whereas many ACL injuries occur in fatigued states. Assessing TTS under fatigued conditions could significantly increase the clinical value of the findings.

Answer 8: This limitation has been added to the Limitations section: “In the present study, jump-landing tasks were performed under non-fatigued conditions, limiting the applicability of findings to real-world sport scenarios where many ACL injuries occur under fatigue. Future studies should assess TTS and related outcomes during fatigued states to better evaluate the practical and clinical relevance of WBV for injury prevention in sport-specific contexts.”

Question 9: In the discussion section, there is a strong emphasis on the improvement in TTS, yet no improvement was observed in SI. SI has higher reliability (ICC: 0.96) compared to TTS; the lack of improvement in SI despite improvement in TTS requires a more precise interpretation: does this TTS improvement truly indicate improved movement quality, or is it merely a temporal change without actual improvement in real control?

Answer 9: We appreciate the reviewer’s comment regarding the apparent discrepancy between TTS and SI outcomes. The intervention group demonstrated reductions in TTS exceeding the minimal detectable change (MDC) previously reported in healthy individuals, suggesting that these improvements likely reflect true enhancements in postural control rather than mere temporal fluctuations.

The lack of significant change in SI, however, can be explained by the fact that SI and TTS assess different aspects of dynamic stability. SI reflects inter-limb symmetry, whereas TTS captures the temporal component of achieving postural stabilization after a jump-landing task. Therefore, participants may achieve faster stabilization without substantial changes in limb symmetry, accounting for the unchanged SI values. We recognize that TTS and SI measure complementary, but not identical, dimensions of postural control, and both should be considered when interpreting the results.

We added this section to the Discussion: “Although SI has higher reliability than TTS, the reductions in TTS exceeding the MDC suggest that these improvements likely reflect genuine enhancements in postural control rather than mere temporal fluctuations, even in the absence of significant changes in SI. This discrepancy can be explained by the fact that SI and TTS assess complementary but distinct aspects of dynamic stability: SI reflects inter-limb symmetry, whereas TTS captures the temporal component of achieving postural stabilization after a jump-landing task. Therefore, participants may achieve faster stabilization without substantial changes in limb symmetry, accounting for the unchanged SI values [20, 32]. Together, these measures provide a more comprehensive understanding of postural control adaptations, and both should be considered when interpreting the results.”

Question 10: Line 93: fix capitalization in “An important methodological limitation” → “an important methodological limitation” if sentence is continuous. Ensure all figures/tables (e.g., force plate setup) include scales and legends. Check and update references (e.g., Slobounov et al., 2006 for age effects), verify DOIs.

Answer 10: Thank you for your comment. Upon checking, the manuscript already uses “…an important methodological limitation…” with “an” in lowercase, so no change is needed. All images include relevant scales and legends. We added DOI for Slobounov et al.

---

## [Decision Letter · Decision Letter 2]

9 Jan 2026

Effect of Whole-Body Vibration Combined With Exercise Therapy on Jump-Landing Stability After ACL Reconstruction: A Randomized Controlled Trial

PONE-D-25-57068R2

Dear Dr. Khanmohammadi,

We’re pleased to inform you that your manuscript has been judged scientifically suitable for publication and will be formally accepted for publication once it meets all outstanding technical requirements.

Kind regards,

Esedullah Akaras

Academic Editor

PLOS One

Additional Editor Comments (optional):

Reviewers' comments:

Reviewer's Responses to Questions

**Comments to the Author**

Reviewer #1: (No Response)

Reviewer #2: All comments have been addressed

2. Is the manuscript technically sound, and do the data support the conclusions?

Reviewer #1: (No Response)

Reviewer #2: Yes

3. Has the statistical analysis been performed appropriately and rigorously?

Reviewer #1: (No Response)

Reviewer #2: Yes

4. Have the authors made all data underlying the findings in their manuscript fully available?

Reviewer #1: (No Response)

Reviewer #2: Yes

5. Is the manuscript presented in an intelligible fashion and written in standard English?

Reviewer #1: (No Response)

Reviewer #2: Yes

Reviewer #1: (No Response)

Reviewer #2: (No Response)

**Do you want your identity to be public for this peer review?** For information about this choice, including consent withdrawal, please see our Privacy Policy

Reviewer #1: No

Reviewer #2: No

---

## [Editor Report · Acceptance letter]

PONE-D-25-57068R2

PLOS One

Dear Dr. Khanmohammadi,

I'm pleased to inform you that your manuscript has been deemed suitable for publication in PLOS One. Congratulations! Your manuscript is now being handed over to our production team.

Kind regards,

on behalf of

Dr. Esedullah Akaras

Academic Editor

PLOS One